# A multi-parameter expansion for the evolution of asymmetric binaries in astrophysical environments

Sayak Datta[1,2]* and Andrea Maselli[1,2]†

**1** Gran Sasso Science Institute (GSSI), I-67100 L'Aquila, Italy
**2** INFN, Laboratori Nazionali del Gran Sasso, I-67100 Assergi, Italy

* sayak.datta@gssi.it , † andrea.maselli@gssi.it

## Abstract

Compact binaries with large mass asymmetries - such as Extreme and Intermediate Mass Ratio Inspirals - are unique probes of the astrophysical environments in which they evolve. Their long-lived and intricate dynamics allow for precise inference of source properties, provided waveform models are accurate enough to capture the full complexity of their orbital evolution. In this work, we develop a multi-parameter formalism, inspired by vacuum perturbation theory, to model asymmetric binaries embedded in general matter distributions with both radial and tangential pressures. In the regime of small deviations from the Schwarzschild metric, relevant to most astrophysical scenarios, the system admits a simplified description, where both metric and fluid perturbations can be cast into wave equations closely related to those of the vacuum case. This framework offers a practical approach to modeling the dynamics and the gravitational wave emission from binaries in realistic matter distributions, and can be modularly integrated with existing results for vacuum sources.

## Contents

Coalescing binaries with large mass asymmetry, i.e., mass ratios $q \ll 1$, represent a novel class of gravitational wave (GW) sources for next-generation detectors, as they remain undetectable by current interferometers. These systems consist of a stellar or an intermediate-mass compact object (the secondary) orbiting a significantly more massive black hole (the primary).

Among these, Extreme Mass Ratio Inspirals (EMRIs), where a primary of mass $\sim (10^6 - 10^8)M_\odot$ pairs with a companion of $\sim (10 - 10^2)M_\odot$, can be observed continuously for tens of thousands of orbits [1]. During this phase, the secondary evolves within a few gravitational radii of the primary before the final plunge, emitting GWs that peak in the millihertz regime—well within LISA's [2] or TianQin's [3] sensitivity range.[1] Intermediate Mass Black Holes (IMBHs), with masses in the range $(10^2 - 10^4)M_\odot$, can form Intermediate Mass Ratio Inspirals (IMRIs) when coupled with either stellar-mass or supermassive black holes (BHs), with mass ratios $q \sim 10^{-4} - 10^{-2}$ [6, 7]. IMRIs have shorter inspirals and less variability than EMRIs [8], emitting GWs across a broad frequency range, from $10^{-3}$ Hz to 10 Hz. This makes them multi-band sources, potentially detectable by mHz [9, 10], decihertz observatories [11], and 3G detectors [12, 13].

As $q$ decreases, the inspiral duration and the number of GW cycles followed by asymmetric binaries increase significantly [14]. These systems spend a substantial portion of their inspiral in a strong-field regime, tracing highly relativistic, eccentric, and off-equatorial trajectories before merging. The combination of such a large number of GW cycles and rich relativistic dynamics is crucial for achieving unprecedented precision in measuring source parameters [1], and advancing the fundamental physics science goals expected by GW observations of these systems [15–18].

Asymmetric binaries have garnered increasing attention as prime sources for probing the astrophysical environments in which they evolve [15]. Indeed, BHs do not exist in isolation; they inhabit diverse environments where particles and fields, potentially of unknown or exotic nature, interact both with each other and with the compact objects. For instance, massive BHs are often surrounded by dark matter halos, which may consist of exotic fields or beyond-standard-model candidates [19]. These surrounding structures can redistribute in the presence of a BH, forming overdensities that influence the binary's orbital dynamics and imprint characteristic signatures on the emitted GW signals [20, 21]. Such signals carry valuable information about changes in the galactic potential and local interactions, such as those arising from dynamical friction [22–32].

Moreover, crowded galactic centers can induce tidal resonances that influence EMRI evolution and reveal nearby stellar-mass object distributions [33]. IMRIs are also expected to form in dense, matter-dominated environments, such as the accretion disks of active galactic nu-

---

[1]Exotic scenarios, such as those involving sub-solar black holes, could allow EMRIs with primaries as light as $10^3 M_\odot$, making them potential targets for third-generation detectors, with GW emission frequencies below 10 Hz [4, 5].

clei [8]. These systems interact with the surrounding gas through effects such as density wakes, gap-opening processes, and tidal torques, leading to complex GW emission patterns [34]. Observing such effects could constrain disk properties and enable multi-messenger analyses via electromagnetic counterparts [35].

Modeling GW emission from asymmetric binaries requires, however, highly accurate waveforms [36]. The self-force (SF) formalism provides the most precise framework to describe such systems, capturing their full evolutionary complexity [14, 37]. In this approach, Einstein field equations are expanded in powers of the mass ratio $q$. The leading-order solution models the secondary as a point particle moving along the geodesics of the primary, while higher-order corrections account for self-interaction and finite-size effects. On the radiation-reaction timescale, the GW phase evolution in the SF expansion follows:

$$\varphi = \frac{\varphi^{(0)}}{q} + \varphi^{(1)} + q\varphi^{(2)} + \cdots, \tag{1}$$

where $\varphi^{(0)}$ and $\varphi^{(1)}$ correspond to the adiabatic (0PA) and post-adiabatic (1PA) contributions, respectively [37]. Phase accuracy at sub-radian levels is needed for precise parameter estimation, requiring calculations up to at least the 1PA order. The leading dissipative effects govern the 0PA phase evolution, while[2] first-order conservative SF and second-order dissipative SF effects contribute to the 1PA phase component $\varphi^{(1)}$. After nearly three decades of effort, recent work has achieved the first implementation of a 1PA waveform [38–40].

Moving beyond vacuum General Relativity presents significant challenges due to the lack of relativistic solutions describing BHs embedded in matter and the complexities introduced by metric-matter couplings. As a result, modeling environmental effects on EMRIs often relies on post-Newtonian approaches [24, 41–46], though fully relativistic descriptions remain key to confidently extract small deviation from vacuum predictions [26, 30, 47–56].

Notable exceptions that provide ab initio background models incorporating non-vacuum contributions include studies investigating how ultra-light scalar fields surrounding massive primaries influence EMRI evolution at leading SF order [57, 58]. A recent study built a relativistic perturbative framework for investigating EMRIs and IMRIs in dense environments, focusing on scalar clouds formed via superradiance around Kerr BHs [59, 60], emphasizing the relevance of spin effects in assessing matter contributions to GW signals

Along with fundamental physics motivations, scalar fields likely provide the most accessible framework for modeling environmental effects. Efforts to model the interaction of asymmetric binaries with generic fluids remain limited due to the complexity of the calculations. A fully relativistic approach, recently developed to model GW emission from EMRIs embedded in spherically symmetric matter distributions [25, 61], using both semi-analytical and fully numerical methods [27, 62–65], revealed a rich and intricate phenomenology arising from a fully relativistic treatment. This model also underscored the significant increase in computational complexity due to matter components and their perturbations. As a result, even at 0PA, generating accurate waveforms across a broad parameter space remains unfeasible.

However, in most astrophysically relevant cases, and in the dynamical regimes of interest for GW detectors, environmental effects are expected to be "small". In this regime, the background geometry of asymmetric binaries is dominated by the BH vacuum spacetime, in which both the companion and the surrounding matter act as perturbations, leading to substantial simplifications.

Following this path, we develop a multi-parameter framework to describe the evolution of asymmetric binaries embedded in generic, low-density environments, modeled via a fluid stress-energy tensor. We adopt a general anisotropic prescription that incorporates both radial

---

[2]Orbital resonances introduce additional corrections at the 0.5PA order [37].

104 and tangential pressure components. Focusing on non-spinning BHs, we solve Einstein equa-
105 tions by computing axial and polar perturbations at first order in the mass ratio. We provide
106 practical, ready-to-use formulas for computing both gravitational and fluid perturbations, as
107 well as the resulting GW emission at the adiabatic order, expressed in terms of environmental
108 parameters and the secondary's orbital trajectory. Throughout this work, we use units in which
109 $G = c = 1$, unless speficied otherwise.

# 1   Field equations and the Multi-parameter expansion

111 Our starting point is the action for generic environmental fields $\vartheta$:

$$S = \int \frac{\sqrt{-g}}{16\pi} d^4x \, \mathcal{R} + S_e[g_{\mu\nu}, \vartheta] + S_p[g_{\mu\nu}, \varphi] \,, \tag{2}$$

112 where the action $S_p$ describes the perturber secondary of mass $m_p$ and its internal matter fields
113 $\varphi$, which can be treated using a skeletonized approach [66], $\mathcal{R}$ is the Ricci scalar, and $g$ the
114 metric determinant. The field equations for $g_{\mu\nu}$, can be derived by varying the total action
115 with respect to the metric, that yields

$$G_{\mu\nu} = 8\pi T^e_{\mu\nu} + 8\pi T^p_{\mu\nu} \,, \tag{3}$$

116 where $G_{\mu\nu}$ is the Einstein operator, and $T^{e,p}_{\mu\nu}$ are the stress-energy tensors related to the envi-
117 ronment and the secondary,

$$T^{e,p}_{\mu\nu} = -\frac{16\pi}{\sqrt{-g}} \frac{\delta\sqrt{-g}\mathcal{L}_{e,p}}{\delta g^{\mu\nu}} \,, \tag{4}$$

118 where $\mathcal{L}_{e,p}$ are the Lagrangian densities associated with the actions $S_{e,p}$. The total energy-
119 momentum tensor satisfies the covariant equation

$$\nabla_\mu T^\mu{}_\nu = \nabla_\mu (T^{e\mu}{}_\nu + T^{p\mu}{}_\nu) = 0 \,. \tag{5}$$

120      We assume the primary is a BH of mass $M$ dressed by a stationary distribution of matter,
121 with a stress-energy tensor for a generic anisotropic fluid[3]:

$$T^e_{\mu\nu} = \rho u_\mu u_\nu + p_r k_\mu k_\nu + p_t \Pi_{\mu\nu} \,, \tag{6}$$

122 where we call $p_t$ and $p_r$ as radial and tangential pressures, $u^\mu$ is the fluid four velocity and
123 $k^\mu$ is a unit space-like radial vector orthogonal to the later, such that $-u_\mu u^\mu = k_\mu k^\mu = 1$ and
124 $u_\mu k^\mu = 0$ [70–72]. The projector on the surface orthogonal to the 4-velocity and $k^\mu$ is given
125 by $\Pi_{\mu\nu} = g_{\mu\nu} + u_\mu u_\nu - k_\mu k_\nu$, with $u^\mu \Pi_{\mu\nu} X^\nu = k^\mu \Pi_{\mu\nu} X^\nu = 0$, for a generic vector $X^\nu$.
126      The secondary BH can be introduced with a perturbative approach, using the mass ratio
127 $q = m_p/M \ll 1$ as parameter of the expansion. In this work we consider linear-order perturba-
128 tions in $q$, which correspond to the leading dissipative contribution in a generic SF expansion
129 of the binary dynamics [14]. In this setup, the secondary evolves along a flow of geodesics
130 driven by the energy and angular momentum fluxes. Higher-order terms, as well as a two-
131 timescale analysis of environmental effects, will be studied elsewhere. The energy momentum
132 of the secondary is given by:

$$T^{p\mu\nu}(x^\alpha) = m_p \int_\gamma u^\mu_p u^\nu_p \frac{\delta^{(4)}(x^\mu - x^\mu_p(\tau))}{\sqrt{-g}} d\tau \,, \tag{7}$$

---

[3]A prescription to describe anisotropic fluids in Newtonian gravity and in General Relativity has been recently proposed in [67–69], aiming to cure certain inconsistencies arising due to Eq. (6) when modeling stellar solutions. Such formalism can in principle be adapted to our approach.

133 where $\gamma$ is the worldline of the compact object, $\tau$ its proper time, and $u_p^\mu(\tau) = dx_p^\mu/d\tau$ its 4–
134 velocity.

135     We introduce a bookkeeping parameter $\epsilon$ to characterize the perturbative nature of the
136 matter distribution, which will later guide the classification of environmental effects. With $\rho$
137 setting the scale of the environmental stress-energy tensor (6), we follow [59] and define $\epsilon$ as
138 the ratio between the environmental and BH densities, $\epsilon = (M_e/L_e^3)/(M/L^3)$, where $M_e$ and
139 $L_e$ are the mass and the scale of the distribution, and $L \sim M$ the BH scale. For instance, in the
140 case of the dark matter configurations considered in [25], one finds $\epsilon = (M_{\text{halo}}/M)/(a_0/L)^3$,
141 with $M_{\text{halo}}$ and $a_0$ denoting the halo mass and its typical size, respectively. In addition to
142 density, the compactness of the matter distribution, defined as $\mathcal{C}_e = M_e/L_e$, is expected to play
143 a central role in determining the behavior of perturbations [25, 27]. Expressing $\epsilon$ in terms of
144 $\mathcal{C}_e$ one obtains $\epsilon \sim \mathcal{C}_e^3(M/M_e)^2$, suggesting that the perturbative treatment remains valid as
145 long as $\mathcal{C}_e \lesssim (M_e/M)^{2/3}$. For example, for typical dark matter halos, with $M_e \sim (10^5 - 10^6)M$,
146 the compactness satisfies $\mathcal{C}_e \ll 1$, ensuring $\epsilon \ll 1$.

147     When $\epsilon \sim \mathcal{O}(1)$, the background metric deviates significantly from the Kerr solution. Con-
148 versely, when $\epsilon \ll \mathcal{O}(1)$, environmental effects can be treated as small perturbations of the
149 vacuum BH background, and the binary dynamics is governed by two small parameters: $\epsilon$ and
150 the mass ratio $q$.

151     In this work, we focus on the latter regime and compute the equations describing metric
152 and matter perturbations by expanding the field equations (3), the covariant conservation of
153 $T^\mu{}_\nu$ (5), and all relevant tensor quantities in powers of $\epsilon$ and $q$. We retain terms up to $\mathcal{O}(\epsilon q)$,
154 such that the metric and stress-energy tensors can be expressed as:

$$g_{\mu\nu} = g_{\mu\nu}^{(0,0)} + q g_{\mu\nu}^{(1,0)} + \epsilon g_{\mu\nu}^{(0,1)} + q\epsilon g_{\mu\nu}^{(1,1)}, \tag{8}$$

$$T_{\mu\nu}^e = \epsilon T_{\mu\nu}^{e\,(0,1)} + q\epsilon T_{\mu\nu}^{e\,(1,1)} \quad , \quad T_{\mu\nu}^p = q T_{\mu\nu}^{p\,(1,0)} + q\epsilon T_{\mu\nu}^{p\,(1,1)}, \tag{9}$$

155 where superscripts $(i, j)$ identify the expansion order $\mathcal{O}(q^i, \epsilon^j)$. In the limit $\epsilon \to 0$ the formal-
156 ism reduces to a particle moving in the Schwarzschild spacetime, with perturbations described
157 by the Regge-Wheeler-Zerilli equations [73–75].

158     **To isolate the various contributions at orders $\epsilon$ and $q$, we expand the nonlinear**
159 **Einstein tensor $G_{\mu\nu}[g_{\alpha\beta}]$ about the background $g_{\alpha\beta}^{(0,0)}$, as in Eq. (8). For a generic pertur-**
160 **bation $h_{\alpha\beta}$, we define the $n$-th variations by**

$$G_{\mu\nu}^{[n]}[h_{\alpha\beta}] = \frac{1}{n!} \frac{d^n}{d\lambda^n} G_{\mu\nu}\left[g_{\alpha\beta}^{(0,0)} + \lambda h_{\alpha\beta}\right]\bigg|_{\lambda=0}. \tag{10}$$

161 **Then**

$$G_{\mu\nu}[g_{\alpha\beta}^{(0,0)} + h_{\alpha\beta}] = G_{\mu\nu}[g^{(0,0)}] + G_{\mu\nu}^{[1]}[h_{\alpha\beta}] + G_{\mu\nu}^{[2]}[h_{\alpha\beta}, h_{\alpha\beta}] + G_{\mu\nu}^{[3]}[h_{\alpha\beta}, h_{\alpha\beta}, h_{\alpha\beta}] + \dots. \tag{11}$$

162 **Inserting the metric expansion (8) into Eq. (11) and keeping terms up to mixed order**
163 $\mathcal{O}(\epsilon q)$ **yields**

$$\begin{aligned} G_{\mu\nu}[g_{\alpha\beta}] = G_{\mu\nu}[g^{(0,0)}] &+ \epsilon\, G_{\mu\nu}^{[1]}[g^{(0,1)}] + q\, G_{\mu\nu}^{[1]}[g^{(1,0)}] \\ &+ \epsilon q\Big(G_{\mu\nu}^{[1]}[g^{(1,1)}] + G_{\mu\nu}^{[2]}[g^{(1,0)}, g^{(0,1)}]\Big) \end{aligned} \tag{12}$$

164 **We assume the background solves the zeroth-order field equations, $G_{\mu\nu}[g^{(0,0)}] = 0$, which**
165 **in Schwarzschild coordinates $x^\mu = (t, r, \theta, \phi)$ gives**

$$g_{\mu\nu}^{(0,0)} = \text{diag}\left(-f, f^{-1}, r^2, r^2\sin^2\theta\right), \qquad f = 1 - \frac{2M}{r}. \tag{13}$$

For clarity, in what follows we absorb the explicit factors of $q$ and $\epsilon$ within each term of the expansion.

The perturbative framework developed above is valid when both the amplitude of the environmental effects and the contribution of the secondary remain small, i.e. for $\epsilon \ll 1$ and $q \ll 1$. Within this regime, nonlinear backreaction on the background geometry is perturbative, and all quantities in Eqs. (8)–(12) can be consistently expanded in powers of these parameters.

Moreover, we can estimate the regime in which nonlinear hydrodynamic effects within the fluid may become relevant by introducing an additional, although Newtonian, physical scale that controls the strength of the local fluid response. In our spherically symmetric configuration, the Bondi–Hoyle–Lyttleton radius $r_B$ [76] provides a diagnostic of the region where the surrounding fluid becomes gravitationally bound to the secondary and nonlinear effects may arise. For orbits at radius $r = x M$, with $x$ the dimensionless orbital separation in units of the primary mass $M$, the ratio $r_B/r \sim q/[x(c_s^2 + v_{\mathrm{rel}}^2)]$ remains well below unity for typical EMRIs ($q \sim 10^{-5}$) whenever either the sound speed $c_s$ or the relative velocity $v_{\mathrm{rel}}$ between the fluid and the secondary exceeds a few $10^{-3}c$, where $c$ is the speed of light [77–79]. This condition is naturally met in warm or hot subsonic flows, ensuring that the fluid response stays in the linear regime and that the point-particle approximation holds.

## 2 Solutions of the multi-parameter expansion

### 2.1 Environmental effects: (0,1) contributions

The $(0,1)$ corrections to the metric tensor satisfy the inhomogeneous equations

$$G^{[1]\mu}{}_{\nu}\big[g^{(0,1)}\big] = 8\pi T^{e\mu}{}_{\nu}{}^{(0,1)} . \tag{14}$$

To determine the components of the environmental stress-energy tensor, we utilize the normalization and orthogonality properties of the fluid four-velocity and the vector $k^\mu$. For a stationary fluid with $u^\mu = (u^t, 0, 0, 0)$ and $k^\mu = (k^t, k^r, 0, 0)$, these conditions lead to

$$u^t = (-g_{tt})^{-1/2}, \quad k^t = 0, \quad k^r = g_{rr}^{-1/2} . \tag{15}$$

Expanding the metric and matter variables in powers of $\epsilon$, we obtain the explicit form of $T^{e\mu}{}_{\nu}{}^{(0,1)}$:

$$T^{e\mu}{}_{\nu}{}^{(0,1)} = \mathrm{diag}(-\rho^{(0,1)}, p_r^{(0,1)}, p_t^{(0,1)}, p_t^{(0,1)}) . \tag{16}$$

For sake of clarity, hereafter we drop the suffix $(0, 1)$ from the background pressure and density functions.

At order $(0, 1)$, we assume the following ansatz for the metric components:

$$g_{\mu\nu}^{(0,1)} = \mathrm{diag}\left(-fH, \frac{2m}{rf^2}, 0, 0\right) , \tag{17}$$

where both $H(r)$ and $m(r)$ are functions of the radial coordinate $r$ only. We focus on asymptotically flat solutions for which the matter variables vanish at the BH horizon $r_h$. This condition fixes $r_h = 2M$, as in the vacuum case, given that $m(r_h) = 0$. At spatial infinity, the functions behave as $H(r \to \infty) = -2M_e/r + \mathcal{O}\left(1/r^2\right)$ and $m(r \to \infty) = M_e + \mathcal{O}(1/r)$, such that

201 $g_{tt}(r \to \infty) = -1 + 2(M + M_e)/r$ , where $M + M_e$ is the total ADM mass of the system, and
202 $M_e$ denotes the mass of the matter distribution.

203 From the $tt$ and $rr$ components of Eq. (15), we derive two ordinary differential equations
204 for $H$ and $m$:

$$\frac{dm}{dr} = 4\pi r^2 \rho \quad , \quad \frac{r^2 f^2}{2}\frac{dH}{dr} = m + 4\pi r^3 f p_r \ . \tag{18}$$

205 Additionally, the energy-momentum covariant derivative at order $(0, 1)$ gives:

$$\frac{dp_r}{dr} = \frac{2}{r}p_t + \frac{(3M - 2r)}{r^2 f}p_r - \frac{M}{r^2 f}\rho \ . \tag{19}$$

206 Equations (18)-(19) alone do not fully determine a solution for the metric and fluid variables.
207 For a given density profile $\rho(r)$, which depends on the specific matter distribution, additional
208 equations are required to close the system. This is typically provided by an equation of state
209 that relates $p_r, p_t$, and $\rho$, **and that we assume to be barotropic**.

210 The background metric $g_{\mu\nu}^{(0,0)} + g_{\mu\nu}^{(0,1)}$ allows for the study of the geodesic properties of
211 both massless and massive particles. For example, the energy and angular momentum per
212 unit mass, $(\mathcal{E}, \mathcal{L})$, of a massive body on a circular orbit of radius $r_p$ are given by:

$$\mathcal{E} = \mathcal{E}^{(0,0)} + \frac{f_p[(1 - 4f_p + 3f_p^2)H_p - 2f_p M H_p']}{\sqrt{2}(f_p - 1)(3f_p - 1)^{3/2}} \ , \tag{20}$$

$$\mathcal{L} = \mathcal{L}^{(0,0)} + \frac{4f_p^2 M^2 H_p'}{(1 - f_p)^{5/2}(3f_p - 1)^{3/2}} \ . \tag{21}$$

213 where the vacuum expressions read:

$$\mathcal{E}^{(0,0)} = \frac{\sqrt{2}f_p}{(3f_p - 1)^{1/2}} \quad , \quad \mathcal{L}^{(0,0)} = \frac{2M}{(4f_p - 3f_p^2 - 1)^{1/2}} \ , \tag{22}$$

214 and $f_p = 1 - 2M/r_p$, $H_p = H(r_p)$, $H_p' = H'(r)|_{r=r_p}$ The corresponding angular frequency of
215 the body up to the linear order in $\epsilon$ is:

$$\Omega_p = \frac{M^{1/2}}{r_p^{3/2}} + \frac{2MH_p + r_p(r_p - 2M)H_p'}{4\sqrt{M}r_p^{3/2}} \ . \tag{23}$$

## 2.2 The motion of the secondary: (1,0)+(1,1) contributions

217 The motion of the secondary generates time dependent perturbations on both the metric and
218 the matter fields, at the linear order in the mass ratio. For technical reasons, that will be clear
219 at the end of this section, we will treat the left-hand side of Einstein equations working with a
220 single background perturbation tensor

$$\delta g_{\mu\nu} = g_{\mu\nu}^{(1,0)} + g_{\mu\nu}^{(1,1)} \ , \tag{24}$$

221 which solves the linearised field's equations:

$$G^\mu{}_\nu[\delta g_{\alpha\beta}] = 8\pi(T^{p\mu}{}_\nu{}^{(1,0)} + T^{e\mu}{}_\nu{}^{(1,1)} + T^{p\mu}{}_\nu{}^{(1,1)}) \ . \tag{25}$$

222 **Since the decoupling of the vacuum $(1, 0)$ and matter $(1, 1)$ sectors is performed at the**
223 **end of the procedure, the operator on the left-hand side of Eq. (25) implicitly includes**
224 **the terms appearing in the expansion (12), namely the linear operators $G^{[1]\mu}{}_\nu[g^{(1,0)}]$ and**
225 **$G^{[1]\mu}{}_\nu[g^{(1,1)}]$, together with the mixed second-order contribution $G^{[2]\mu}{}_\nu[g_{\alpha\beta}^{(1,0)}, g_{\alpha\beta}^{(0,1)}]$.**

Given the symmetry of the background, metric perturbations can be separated into the usual families of axial ($A$) and polar ($P$) components [73–75]:

$$\delta g_{\mu\nu}(x^\alpha) = \delta g_{\mu\nu}^A(x^\alpha) + \delta g_{\mu\nu}^P(x^\alpha) \,. \tag{26}$$

Axial and polar modes change sign as $(-1)^{\ell+1}$ and $(-1)^\ell$ under the coordinate inversion $(\theta \to \pi - \theta, \phi \to \phi + \pi)$, respectively. The two classes of perturbations decouple, and can be treated independently. We can expand $g_{\mu\nu}^A(x^\alpha)$ and $g_{\mu\nu}^P(x^\alpha)$ in a complete set of tensor harmonics, such that:

$$\delta g_{\mu\nu}^A = \sum_{\ell,m} \frac{\sqrt{2\lambda}}{r}\left[ ih_{1,\ell m}(t,r)\mathbf{c}_{\ell m}(\theta,\phi) - h_{0,\ell m}(t,r)\mathbf{c}^0{}_{\ell m}(\theta,\phi) + \frac{\sqrt{\Lambda}}{r}h_{2,\ell m}(t,r)\mathbf{d}_{\ell m}(\theta,\phi) \right] \,, \tag{27}$$

$$\delta g_{\mu\nu}^P = \sum_{\ell,m}\Big[ -g_{tt}H_{0,\ell m}(t,r)\mathbf{a}^0{}_{\ell m}(\theta,\phi) - i\sqrt{2}H_{1,\ell m}(t,r)\mathbf{a}^1{}_{\ell m}(\theta,\phi) - \frac{i}{r}\sqrt{2\lambda}\eta_{0,\ell m}(t,r)\mathbf{b}^0{}_{\ell m}(\theta,\phi)$$

$$+ \frac{\sqrt{2\lambda}}{r}\eta_{1,\ell m}(t,r)\mathbf{b}_{\ell m}(\theta,\phi) + g_{rr}H_{2,\ell m}(t,r)\mathbf{a}_{\ell m}(\theta,\phi) + \sqrt{\Lambda\lambda}G_{\ell m}(t,r)\mathbf{f}_{\ell m}(\theta,\phi)$$

$$+ \left(\sqrt{2}K_{\ell m}(t,r) - \frac{\lambda}{\sqrt{2}}G_{\ell m}(t,r)\right)\mathbf{g}_{\ell m}(\theta,\phi) \Big] \,, \tag{28}$$

where $\lambda = \ell(\ell+1)$, $\Lambda = (\ell+2)(\ell-1)/2$, and the sum over the multipolar indices $(\ell,m)$ runs from $\ell = 0,\ldots,\infty$ and $m = -\ell,\ldots,\ell$. The ten basis components $\{\mathbf{c}_{\mu\nu}^{\ell m}, \mathbf{c}_{\mu\nu}^{0\ell m} \ldots \mathbf{g}_{\mu\nu}^{\ell m}\}$ depend on the spherical harmonics $Y_{\ell m}(\theta,\phi)$ and their derivatives (see e.g. Appendix A of [80] for their explicit expression). Among the ten unknown functions $\{h_{1\ell m} \ldots K_{\ell m}\}$, the axial term $h_{2\ell m}$ and the three polar components $\{\eta_{0\ell m}, \eta_{1\ell m}, G_{\ell m}\}$ can be set to zero by adopting the Regge-Wheeler-Zerilli gauge, such that the metric satisfy

$$\delta g_{\theta\phi} = 0 \,, \quad \delta g_{\phi\phi} = \delta g_{\theta\theta}\sin^2\theta \,,$$
$$\partial_\phi(\delta g_{t\phi}/\sin\theta) + \partial_\theta(\delta g_{t\theta}/\sin\theta) = 0 \,,$$
$$\partial_\phi(\delta g_{r\phi}/\sin\theta) + \partial_\theta(\delta g_{r\theta}/\sin\theta) = 0 \,. \tag{29}$$

Similarly to the metric perturbations, we decompose the particle stress-energy tensor in the basis of tensor harmonics:

$$T_{\mu\nu}^{p\,(1,0)} = \sum_{\ell,m}\Big[ \mathcal{A}_{\ell m}^{0(1,0)}\mathbf{a}^0{}_{\ell m}(\theta,\phi) + \mathcal{A}_{\ell m}^{1(1,0)}\mathbf{a}^1{}_{\ell m}(\theta,\phi) + \mathcal{A}_{\ell m}^{(1,0)}\mathbf{a}_{\ell m}(\theta,\phi) + \mathcal{B}_{\ell m}^{0(1,0)}\mathbf{b}^0{}_{\ell m}(\theta,\phi)$$

$$+ \mathcal{B}_{\ell m}^{(1,0)}\mathbf{b}_{\ell m}(\theta,\phi) + \mathcal{Q}_{\ell m}^{(1,0)}\mathbf{c}_{\ell m}(\theta,\phi) + \mathcal{Q}_{\ell m}^{0(1,0)}\mathbf{c}^0{}_{\ell m}(\theta,\phi) + \mathcal{D}_{\ell m}^{(1,0)}\mathbf{d}_{\ell m}(\theta,\phi)$$

$$+ \mathcal{G}_{\ell m}^{(1,0)}\mathbf{g}_{\ell m}(\theta,\phi) + \mathcal{F}_{\ell m}^{(1,0)}\mathbf{f}_{\ell m}(\theta,\phi) \Big] \,. \tag{30}$$

The specific form of the coefficients $\{\mathcal{A}_{\ell m}^{0(1,0)}, \ldots \mathcal{F}_{\ell m}^{(1,0)}\}$ depends on the secondary orbital configurations. Finally, the form $T_{\mu\nu}^{p\,(1,1)}$ can be constructed using the same ansatz of Eqs. (30), and replacing the functions with the correct order of the expansion, e.g. $\mathcal{Q}_{\ell m}^{(1,0)} \to \mathcal{Q}_{\ell m}^{(1,1)}$ (see Appendix C for further details).

### 2.2.1 Environmental effects in the presence of the secondary: (1,1) matter decompositions

The last piece of the multi-parameter expansion is given by the $(1,1)$ perturbations of matter energy-momentum tensor, $T^{e\mu}{}_\nu^{(1,1)}$. The covariant equations (5) are determined, at this order, by three contributions:

$$\nabla_\mu[g_{\alpha\beta}^{(0,0)}](T^{e\mu}{}_\nu^{(1,1)} + T^{p\mu}{}_\nu^{(1,1)}) + \nabla_\mu[g_{\alpha\beta}^{(1,0)}]T^{e\mu}{}_\nu^{(0,1)} + \nabla_\mu[g_{\alpha\beta}^{(0,1)}]T^{p\mu}{}_\nu^{(1,0)} = 0 \,, \tag{31}$$

where we identify with $\nabla_\mu[g_{\alpha\beta}^{(m,n)}]$ the covariant derivative depending on the metric at the $(i, j)$ order. The $(1, 1)$ contributions to the matter stress-energy tensor depend on the energy density and the pressure perturbations:

$$\rho = \rho(r) + \rho^{(1,1)}(t, r, \theta, \phi) \, , \tag{32}$$

$$p = p_r(r) + p_r^{(1,1)}(t, r, \theta, \phi) \, , \tag{33}$$

$$p_t = p_t(r) + p_t^{(1,1)}(t, r, \theta, \phi) \, . \tag{34}$$

We exploit again the symmetry of the background to separate angular and time-radial variables. We expand fluid variables in terms of standard spherical harmonics

$$\rho^{(1,1)} = \sum_{\ell,m} \rho_{\ell m}^{(1,1)}(t, r) Y_{\ell m}(\theta, \phi) \, , \tag{35}$$

$$p_r^{(1,1)} = \sum_{\ell,m} p_{r,\ell m}^{(1,1)}(t, r) Y_{\ell m}(\theta, \phi), \tag{36}$$

$$p_t^{(1,1)} = \sum_{\ell,m} p_{t,\ell m}^{(1,1)}(t, r) Y_{\ell m}(\theta, \phi) \, . \tag{37}$$

Moreover, pressure and density perturbations are linked by an equation of state[4], such that:

$$p_{r,\ell m}^{(1,1)} = c_{r,\ell m}^2(r) \rho_{\ell m}^{(1,1)} \quad , \quad p_{t,\ell m}^{(1,1)} = c_{t,\ell m}^2(r) \rho_{\ell m}^{(1,1)} \, , \tag{38}$$

where the tangential ($c_{t,\ell m}^2$) and the radial ($c_{r,\ell m}^2$) sound speeds are in general not constant, and are functions of the radial coordinate (See Ref. [81] for specific examples).

Perturbations of the fluid velocity $u^\mu$ and $k^\mu$ can be written in terms of vector harmonics [82]. Given the form of the matter stress-energy tensor in Eq. (6) and that, to leading order, the energy and pressure variables are $\mathcal{O}(\epsilon)$, we only need terms of the order $u^{\mu(1,0)}$ and $k^{\mu(1,0)}$ to determine $T^{e\mu}{}_\nu^{(1,1)}$. The normalization of the 4-velocity reduces the independent component of the perturbations to three unknown functions. The explicit form of $u^{\mu(1,0)}$ and $k^{\mu(1,0)}$ is given by:

$$u^{t(1,0)} = \frac{1}{2\sqrt{f}} \sum_{\ell,m} H_{0,\ell m}^{(1,0)}(t, r) Y_{\ell m}(\theta, \phi) \, , \tag{39}$$

$$u^{r(1,0)} = \frac{f^{3/2}}{4\pi} \sum_{\ell,m} W_{\ell m}^{(1,0)}(t, r) Y_{\ell m}(\theta, \phi) \, , \tag{40}$$

$$u^{\theta(1,0)} = \frac{\sqrt{f}}{4\pi r^2} \sum_{\ell,m} \left[ V_{\ell m}^{(1,0)}(t, r) \partial_\theta - \frac{U_{\ell m}^{(1,0)}(t, r)}{\sin\theta} \partial_\phi \right] Y_{\ell m}(\theta, \phi) \, , \tag{41}$$

$$u^{\phi(1,0)} = \frac{\sqrt{f}}{4\pi r^2 \sin^2\theta} \sum_{\ell,m} \left[ V_{\ell m}^{(1,0)}(t, r) \partial_\phi + U_{\ell m}^{(1,0)}(t, r) \sin\theta \, \partial_\theta \right] Y_{\ell m}(\theta, \phi) \, . \tag{42}$$

The form of $k^{r(1,0)}$ and $k^{t(1,0)}$ can be found using nomalisation and orthogonality conditions.

# 3 Perturbation equations

The procedure for determining axial and polar perturbations closely follows the vacuum case, which has been extensively studied in the literature since the seminal works of Regge and

---

[4]**Note that, since the physical properties of matter are not altered by linear perturbations, the underlying equation of state is assumed to remain unchanged.**

Wheeler [73, 74] and Zerilli [75]. In this section, we revisit the key steps for deriving the master equations governing the evolution of $\delta g_{\mu\nu}^{(A,P)}$, and for isolating the contributions arising from the $(1,0)$ and $(1,1)$ terms. We refer the reader to Appendix B for further details on our initial assumption of working with a single metric perturbation in $q$, and on the decoupling between vacuum and matter components. We present most of the equations in a compact form, emphasizing their functional dependence on the metric and fluid perturbations. The full explicit expressions are provided in the accompanying `Mathematica` supplementary file [83].

## 3.1 $\ell \geq 2$ axial modes

In the axial case, we use the $\theta\theta$ and $\phi\phi$ components of Eqs. (25) to express the time derivative $\partial_t h_{0\ell m}$ as a function of $h_{1,\ell m}$ and $\partial_r h_{1,\ell m}$. Substituting the latter into the $r\theta$ component of Einstein equations and introducing the master variable $\bar{\phi}_{\ell m} = -h_{1,\ell m}/r(-g_{tt}/g_{rr})^{1/2}$, we obtain a single, second-order partial differential equation of the form:

$$(-g_{tt}/g_{rr})\partial_r^2 \bar{\phi}_{\ell m} - \partial_t^2 \bar{\phi}_{\ell m} + a_1 \partial_r \bar{\phi}_{\ell m} + a_2 \bar{\phi}_{\ell m} = S_{\ell m} , \qquad (43)$$

where $a_{1,2}$ depend only on $(0,0)$ and $(0,1)$ quantities. The source term $S_{\ell m}$ contains contributions from the particle's stress-energy tensor and the background fluid variables. We now introduce a new master function:

$$\phi_{\ell m}(t,r) = \sqrt{Z(r)}\, \bar{\phi}_{\ell m}(t,r) , \qquad (44)$$

where $Z = f^{-1}(-g_{tt}/g_{rr})^{1/2}$. For environmental effects that can be treated as small perturbations of the Schwarzschild metric, as considered here, we write $Z(r) = 1 + \delta Z(r)$, where $\delta Z(r)$ is of order $\mathcal{O}(\epsilon)$:

$$\delta Z(r) = \frac{H(r)}{2} - \frac{m(r)}{rf} . \qquad (45)$$

In terms of the new field $\phi_{\ell m}$, Eq. (43) becomes:

$$f\,\partial_r(f\,\partial_r\phi_{\ell m}) + (1 - \delta Z)\partial_t^2 \phi_{\ell m} + a_3 \phi_{\ell m} = \bar{S}_{\ell m} , \qquad (46)$$

At this point we can decompose the perturbation into vacuum and matter components, i.e., $\phi_{\ell m} = \phi_{\ell m}^{(0,0)} + \phi_{\ell m}^{(1,1)}$. Furthermore, by introducing the usual tortoise coordinate $r_\star$, such that $\partial_{r_*} = f\partial_r$, we can eliminate the first radial derivative of the metric perturbations, obtaining the following wave equations:

$$[\partial_{r_\star}^2 - \partial_t^2 - V^A]\phi_{\ell m}^{(1,0)}(t,r) = S_{\ell m}^{A(1,0)}(t,r) , \qquad (47)$$

$$[\partial_{r_\star}^2 - \partial_t^2 - V^A]\phi_{\ell m}^{(1,1)}(t,r) = S_{\ell m}^{A(1,1)}(t,r) . \qquad (48)$$

Thus, at linear order in $\mathcal{O}(\epsilon)$, the axial perturbation problem reduces to solving two wave equations, with the same scattering potential, which matches the Regge–Wheeler expression for the vacuum case:

$$V^A = f\left(\frac{\ell(\ell+1)}{r^2} - \frac{6M}{r^3}\right) . \qquad (49)$$

The source $S_{\ell m}^{A(1,0)}(t,r)$ only depends on the coefficients of $T_{\mu\nu}^{p(1,0)}$ in Eq. (30). The source $S_{\ell m}^{A(1,1)}$ is proportional to $T_{\mu\nu}^{p(1,1)}$, and contains contributions from the vacuum master function $\phi_{\ell m}^{(1,0)}$, multiplied by the matter density and pressure. Once Eqs. (47)–(48) are solved, we can use Eq. (45) to obtain the $(1,0)$ and $(1,1)$ components of $\phi_{\ell m}$, and consequently the expansion for the metric functions $h_{1,\ell m} = h_{1,\ell m}^{(1,0)} + h_{1,\ell m}^{(1,1)}$ and $h_{0,\ell m} = h_{0,\ell m}^{(1,0)} + h_{0,\ell m}^{(1,1)}$. Their explicit expressions are given in Appendix A.

Finally, the velocity perturbation $U_{\ell m}^{(1,0)}$ can be derived from the $t\theta$ component of Einstein equations, which yields an algebraic relation between this quantity and the metric variables:

$$\partial_t U_{\ell m}^{(1,0)}(t,r) = -\frac{4\pi\partial_t h_{0,\ell m}^{(1,0)}}{f} + \frac{4\pi(3-2r)(\kappa_r - \kappa_t)h_{1,\ell m}^{(1,0)}}{r^2\kappa_t} + S_{\ell m}^U(t,r) \,, \tag{50}$$

with $S_{\ell m}^U(t,r)$ depending on the point particle motion, $\kappa_t = \rho(r)+p_t(r)$, and $\kappa_r = \rho(r)+p_r(r)$. **Axial perturbations do not couple, at this order, to the energy density or pressure perturbations because of parity considerations.**

### 3.1.1 The frequency domain solution

In the frequency domain, Eqs. (47)–(48) reduce to two ordinary differential equations in the radial coordinate:

$$[\partial_{r_\star}^2 + \omega^2 - V^A]\phi_{\ell m}^{(1,0)}(\omega,r) = S_{\ell m}^{A(1,0)}(\omega,r) \,, \tag{51}$$

$$[\partial_{r_\star}^2 + \omega^2 - V^A]\phi_{\ell m}^{(1,1)}(\omega,r) = S_{\ell m}^{A(1,1)}(\omega,r) \,, \tag{52}$$

where, for a generic function $X(t,r)$:

$$X(\omega,r) = \frac{1}{2\pi}\int_{-\infty}^{+\infty} e^{i\omega t}X(t,r)\,dt \quad, \quad X(t,r) = \int_{-\infty}^{+\infty} e^{-i\omega t}X(\omega,r)\,d\omega \,. \tag{53}$$

Equations (51)–(52) can be solved using a Green's function approach. We first solve the associated homogeneous equations with purely ingoing (-) boundary conditions at the horizon and purely outgoing (+) conditions at infinity:

$$\phi_{\ell m}^{(1,0)(-)} \sim \begin{cases} e^{-i\omega r_\star} & r_\star \to -\infty \\ A_{in}e^{-i\omega r_\star} + A_{\text{out}}e^{i\omega r_\star} & r_\star \to +\infty \end{cases} \,, \tag{54}$$

$$\phi_{\ell m}^{(1,0)(+)} \sim \begin{cases} e^{i\omega r_\star} & r_\star \to +\infty \\ B_{\text{in}}e^{-i\omega r_\star} + B_{\text{out}}e^{i\omega r_\star} & r_\star \to -\infty \end{cases} \,, \tag{55}$$

Note that the homogeneous equation is identical for both the $(1,0)$ and $(1,1)$ components, and hence needs to be solved only once. The full solution is obtained by integrating $\phi_{\ell m}^{(1,0)(\pm)}$ over the source term:

$$\phi_{\ell m}^{(1,0)} = C^+\phi_{\ell m}^{(1,0)(-)} + C^-\phi_{\ell m}^{(1,0)(+)} \,, \tag{56}$$

with coefficients given by:

$$C^+ = \int_{-\infty}^{r_\star} \frac{\phi_{\ell m}^{(1,0)(-)}(r_\star')S_{\ell m}^{(1,0)}(r_\star')}{\mathcal{W}_{\ell m}(r_\star')}\,dr_\star' \quad, \quad C^- = \int_{r_\star}^{+\infty} \frac{\phi_{\ell m}^{(1,0)(+)}(r_\star')S_{\ell m}^{(1,0)}(r_\star')}{\mathcal{W}_{\ell m}(r_\star')}\,dr_\star' \,, \tag{57}$$

where $\mathcal{W}_{\ell m}$ is the constant Wronskian of the homogeneous solutions:

$$\mathcal{W}_{\ell m}(r_\star) = f\,\partial_r\phi_{\ell m}^{(1,0)(+)}\phi_{\ell m}^{(1,0)(-)} - f\,\partial_r\phi_{\ell m}^{(1,0)(-)}\phi_{\ell m}^{(1,0)(+)} \,. \tag{58}$$

The solution for $\phi_{\ell m}^{(1,1)}$ has the same form as Eq. (56), with the substitution $S_{\ell m}^{(1,0)} \to S_{\ell m}^{(1,1)}$ in the $C^\pm$ coefficients.

For circular orbits the calculation of $C^\pm$ greatly simplifies. In this case the source term can be written as function of Dirac's delta and it's first derivative:

$$S_{\ell m}^{(1,0)} = D(r,r_p)\delta(r-r_p) + G(r,r_p)\delta'(r-r_p) \,, \tag{59}$$

where $r_p$ is the secondary orbital radius, and the functions $D, G$ can be determined from the coefficients of $T_{\mu\nu}^{p(1,0)}$ (and of $T_{\mu\nu}^{p(1,1)}$ for the matter contribution). Integration in Eqs. (57)can be performed analytically such that

$$C^+ = \mathcal{C}^+ \Theta(r - r_p) \quad , \quad C^- = \mathcal{C}^- \Theta(r_p - r) \,, \tag{60}$$

where

$$\mathcal{C}^\pm = \frac{\phi_{\ell m}^{(1,0)(\mp)}(r_p)D(r_p)}{f_p \mathcal{W}} - \frac{d}{dr}\left[\frac{\phi_{\ell m}^{(1,0)(\mp)}(r_p)G(r_p)}{\mathcal{W}f(r)}\right]_{r=r_p} . \tag{61}$$

## 3.2   $\ell \geq 2$ polar modes

Perturbations in the polar sector are characterized by seven variables: four metric components $(H_{0,\ell m}, H_{1,\ell m}, H_{2,\ell m}, K_{\ell m})$, two components of the fluid velocity perturbation $(V_{\ell m}^{(1,0)}, W_{\ell m}^{(1,0)})$, and the density perturbation $\rho^{(1,1)}$. Despite this complexity, the dimensionality of the system can be significantly reduced.

The $\theta\phi$ component of Eqs. (25) allows us to express $H_{2,\ell m}$ in terms of $H_{0,\ell m}$. Furthermore, the $rr$, $tr$, and $t\theta$ components of Einstein equations can be used to eliminate the time derivative of $H_{0,\ell m}$, yielding two coupled differential equations[5] that depend only on the metric functions $K_{\ell m}$ and $H_{1,\ell m}$, along with the fluid perturbations, and take the following form:

$$(b_1 + b_2\partial_r)H_{1,\ell m} + (b_3\partial_t + b_4\partial_{tr}^2)K_{\ell m} + b_5 V_{\ell m}^{(1,0)} + b_6 W_{\ell m}^{(1,0)} = S_{\ell m}^H \,, \tag{62}$$

$$(c_1 + c_2\partial_r + c_3\partial_{rr}^2 + c_4\partial_{tt}^2)H_{1,\ell m} + (c_4\partial_t + c_5\partial_{tr}^2)K_{\ell m} + (c_6 + c_7\partial_r)V_{\ell m}^{(1,0)} + c_8 W_{\ell m}^{(1,0)} = S_{\ell m}^K \,, \tag{63}$$

From the time component of the covariant derivative of the stress-energy tensor, we obtain an equation for $\rho_{\ell m}^{(1,1)}$:

$$d_1\partial_t\rho_{\ell m}^{(1,1)} + d_2\partial_t K_{\ell m} + (d_3 + d_4\partial_r)H_{1,\ell m} + (d_5 + d_6\partial_r)W_{\ell m}^{(1,0)} + d_7 V_{\ell m}^{(1,0)} = \mathcal{J}_{\ell m}^\rho \,. \tag{64}$$

The coefficients $(b_i, c_i, d_i)$ appearing in Eqs. (62)–(64) contain background quantities and depend only on $r$. **Solving Eq. (64) allows to determine $\rho_{\ell m}^{(1,1)}$, and hence $(p_{r,\ell m}^{(1,1)}, p_{t,\ell m}^{(1,1)})$, as a function of background quantities and of the vacuum solution through Eq. (38).**

Finally, from $\nabla_\mu T^{\mu\theta} = 0$ and $\nabla_\mu T^{\mu r} = 0$, we obtain two first-order equations in time for $\partial_t V_{\ell m}^{(1,0)}$ and $\partial_t W_{\ell m}^{(1,0)}$.

We now reduce the coupled system for $H_{1,\ell m}$ and $K_{\ell m}$ to a single master equation for the metric perturbation, following the strategy introduced by Zerilli [75,84], and isolate its $(0, 1)$ and $(1, 1)$ components. We first introduce the new functions $\bar{\chi}_{\ell m}(t, r)$ and $\bar{R}_{\ell m}(t, r)$:

$$\partial_t K_{\ell m}(t, r) = \alpha\bar{\chi}_{\ell m}(t, r) + \beta\bar{R}_{\ell m}(t, r) \quad , \quad H_{1,\ell m}(t, r) = \gamma\bar{\chi}_{\ell m}(t, r) + \delta\bar{R}_{\ell m}(t, r) \,, \tag{65}$$

As in the axial sector, we introduce the scaling function $Z(r)$ such that $\chi_{\ell m} = \sqrt{Z}\bar{\chi}_{\ell m}$ and $R_{\ell m} = \sqrt{Z}\bar{R}_{\ell m}$. The coefficients $(\alpha, \beta, \gamma, \delta)$, which depend only on $r$, are fixed by requiring that $\chi_{\ell m}$ and $R_{\ell m}$ satisfy Zerilli-like equations of the form:

$$f\,\partial_r[f\,\partial_r\chi_{\ell m}] + (\mathcal{V}^P - \partial_t^2)\chi_{\ell m} = \mathcal{S}_{\ell m}^P \quad , \quad f\,\partial_r\chi_{\ell m} - R_{\ell m} = \mathcal{J}_{\ell m}^P \,, \tag{66}$$

---

[5]These algebraic manipulations also introduce a third-order time derivative of $K_{\ell m}$, which can be removed using the $t\phi$ component of Einstein equations.

for some scattering potential $\mathcal{V}^P$. At this stage, and for readability, we collectively include in the source terms $\mathcal{S}_{\ell m}^P$ and $\mathcal{J}_{\ell m}^P$ all contributions proportional to the secondary orbital configuration and fluid perturbations. Their explicit forms will be given later.

The coefficients that 'diagonalize' the problem coincide with those originally found by Zerilli [75, 84]. At this point, all metric perturbations can be expanded in the two-parameter scheme, e.g., $\chi_{\ell m} = \chi_{\ell m}^{(0,1)} + \chi_{\ell m}^{(1,1)}$. As a result, Eqs. (62)–(63) reduce to:

$$\partial_{r_\star}^2 \chi_{\ell m}^{(1,0)} + (V^P - \partial_t^2)\chi_{\ell m}^{(1,0)} = S_{\ell m}^{P(1,0)} , \tag{67}$$

$$\partial_{r_\star}^2 \chi_{\ell m}^{(1,1)} + (V^P - \partial_t^2)\chi_{\ell m}^{(1,1)} + (z_1 + z_2 f \partial_r)V_{\ell m}^{(1,0)} + (z_3 + z_4 f \partial_r)W_{\ell m}^{(1,0)} = S_{\ell m}^{P(1,1)} . \tag{68}$$

The scattering potential for both the $(1,0)$ and $(1,1)$ equations coincides and is given by the well-known vacuum result:

$$V^P = -\frac{2f}{r^3}\frac{9M^3 + 9\Lambda M^2 r + 3\Lambda^2 M r^2 + \Lambda^2(\Lambda+1)r^3}{(3M + r\Lambda)^2}. \tag{69}$$

The source term $S_{\ell m}^{P(1,1)}$ is proportional to $\chi_{\ell m}^{(1,0)}$ and to the components of $T_{\mu\nu}^{p,(1,1)}$, while the coefficients $z_{1,2,3,4}$ depend only on the background pressure and density. The density perturbation enters the equation for $\chi_{\ell m}^{(1,1)}$ via the fluid velocities, which are determined by:

$$\kappa_t \partial_t V_{\ell m}^{(1,0)} + 4\pi c_{t,\ell m}^2 \rho_{\ell m}^{(1,1)} = S_{\ell m}^V , \tag{70}$$

$$\kappa_r \partial_t W_{\ell m}^{(1,0)} + (w_1 + w_2 f \partial_r)\rho_{\ell m}^{(1,1)} = S_{\ell m}^W , \tag{71}$$

where $\kappa_t = \rho(r) + p_t(r)$ and $\kappa_r = \rho(r) + p_r(r)$. Finally, using Eqs. (67) and (70)–(71), we can simplify the master equation for $\rho_{\ell m}^{(1,1)}$. Taking the time derivative of Eq. (64) yields:

$$(\partial_{r_\star}^2 - c_{r,\ell m}^{-2}\partial_t^2 + V^\rho + \gamma_1 \partial_{r_\star})\rho_{\ell m}^{(1,1)} = S_{\ell m}^\rho . \tag{72}$$

The coefficients $(w_1, w_2)$ involve combinations of $p_{t,r}(r)$ and $\rho(r)$, while $V^\rho$ and $\gamma_1$ depend only on the sound speeds. Along with the particle motion, the sources $S_{\ell m}^{V,W,\rho}$ depend on the vacuum solutions $\chi_{\ell m}^{(1,0)}$.

Note that Eq. (72) is decoupled from the $(1,1)$ metric perturbations and can be solved once the vacuum solution is known. This allows to determine $V_{\ell m}^{(1,0)}$ and $W_{\ell m}^{(1,0)}$ via Eqs. (70)-(71). These quantities can then be used to fully solve the polar sector and obtain $\chi_{\ell m}^{(1,1)}$ through Eq. (68). The metric components can subsequently be reconstructed using the expressions in Appendix A.

We also briefly comment on the structure of the polar sector in the frequency domain. Although the equations remain too lengthy to present explicitly, the formulation simplifies significantly. In this case, the velocity perturbations, given by Eqs. (70)-(71), reduce to algebraic relations and can be eliminated from the wave equation for $\chi_{\ell m}^{(1,1)}$, which can then be determined once a solution for $\rho_{\ell m}^{(1,1)}$ is obtained using the Green function approach already discussed for the axial sector.

## 3.3 $\ell = 0$ modes

For the sake of completeness, we complement the previous calculations with the treatment of the $\ell = 0$ and $\ell = 1$ modes, which do not contribute to gravitational radiation.

378      Fo $\ell = m = 0$, only polar perturbations are excited. In this case we adopt the so called
379 Zerilli gauge, which allows us to set $H_{1,00} = K_{00} = 0$ [84, 85]. Decomposing the remaining
380 metric functions $H_{2,00}$ and $H_{0,00}$ into vacuum and matter components, we obtain

$$\partial_r H_{0,00}^{(1,0)} = -\frac{H_{2,00}^{(1,0)}}{rf} - 8\pi r A_{00}^{(1,0)} \, , \tag{73}$$

$$\partial_r H_{2,00}^{(1,0)} = -\frac{H_{2,00}^{(1,0)}}{rf} + \frac{8\pi r}{f^2} A_{00}^{0(1,0)} \, , \tag{74}$$

381 which coincide with the standard results derived in the vacuum case [86], and

$$\partial_r H_{0,00}^{(1,1)} = -\frac{8\pi r c_r^2 \rho_{00}^{(1,1)}}{f} - \frac{2}{f^2 r^2}\left(4\pi f r^3 p_r + m\right) H_{2,00}^{(1,0)} - \frac{H_{2,00}^{(1,1)}}{fr} - 8\pi r \mathcal{A}_{00}^{(1,1)} \, , \tag{75}$$

$$\begin{aligned}\partial_r H_{2,00}^{(1,1)} = &-\frac{H_{2,00}^{(1,1)}}{fr} + \frac{8\pi r}{f}\rho_{00}^{(1,1)} + \frac{8\pi r}{f^2}\mathcal{A}_{00}^{(0)(1,1)} + \frac{2}{f^2 r^2}H_{2,00}^{(1,0)}\left(4\pi f r^3\rho - m\right) \\ &- \frac{8\pi}{f^3}\mathcal{A}_{00}^{0(1,0)}(frH - 2m) \, . \end{aligned} \tag{76}$$

382      Moreover, an algebraic equation for $W_{00}^{(1,0)}$ can be obtained from the $tr$ component of
383 Einstein equations:

$$\kappa_r W_{00}^{(1,0)} = \frac{2i\sqrt{2}\pi \mathcal{A}_{00}^{1(1,1)}}{f} - \frac{\partial_t H_{2,00}^{(1,1)}}{2fr} \, , \tag{77}$$

384 Finally, substituting the above into the $\theta\theta$ component of Eqs. (25), we obtain a master equation
385 for $\rho_{00}^{(1,1)}$:

$$\left(\partial_{r_\star}^2 - c_{r,00}^{-2}\partial_t^2 + V_{\ell=0}^\rho + \gamma_{1,\ell=0}\partial_{r_\star}\right)\rho_{00}^{(1,1)} = S_{00}^\rho \, . \tag{78}$$

386 The source term $S_{00}^\rho$ depends on the vacuum solution $H_{2,00}^{(1,0)}$ and on the secondary orbital
387 trajectory, while the potential $V_{\ell=0}^p$ and the coefficient $\gamma_{1,\ell=0}$ contain terms proportional to
388 the radial sound speed. As for the $\ell \geq 2$ modes, Eq. (78) is decoupled from the $(1,1)$ metric
389 perturbations and is entirely determined by the vacuum component. Once solved, one can
390 determine $W_{00}^{(1,0)}$ via Eq. (77), and subsequently reconstruct $H_{0,00}$ and $H_{2,00}$.

## 391   3.4   $\ell = 1$ modes

392 For $\ell = 1$, both axial and polar modes are present. In the axial sector, the Zerilli gauge is
393 implemented by setting $h_{0,1m} = 0$, leaving $h_{1,1m}$ as the only nonvanishing metric component
394 to be determined [75]. The field equations for the $(1,0)$ axial perturbation take the form:

$$\partial_t^2 h_{1,1m}^{(1,0)} = -rf\, 8i\pi\, \mathcal{Q}_{1m}^{(1,0)} \, , \tag{79}$$

$$\frac{2}{r^2}\partial_t h_{1,1m}^{(1,0)} + \frac{1}{r}\partial_{t,r}^2 h_{1,1m}^{(1,0)} + \frac{8\pi}{f}\mathcal{Q}_{1m}^{0(1,0)} = 0 \, . \tag{80}$$

395 At the $(1,1)$ order we have for the metric perturbation

$$\partial_t^2 h_{1,1m}^{(1,1)} - [H\partial_t^2 - 16\pi f(p_t - p_r)]h_{1,1m}^{(1,0)} + 8\pi i f r \mathcal{Q}_{1m}^{(1,1)} = 0 \, , \tag{81}$$

$$\left(f\partial_{t,r}^2 + \frac{2f}{r}\partial_t\right)h_{1,1m}^{(1,1)} + 4f\kappa_t U_{1m}^{(1,0)} + 8\pi r \mathcal{Q}_{1m}^{(0)(1,1)} - \frac{4}{r^2}\left(\pi r^3\kappa_r + m + \frac{rm}{2}\partial_r\right)\partial_t h_{1,1m}^{(1,0)} = 0 \, . \tag{82}$$

Finally, an equation for $\partial_t U_{1m}^{(1,0)}$ can be derived from $\theta$ component of the covariant divergence $\nabla_\mu T^{\mu\theta} = 0$.

In the polar sector, we fix the Zerilli gauge by setting $K_{1m} = 0$, so that the remaining metric perturbations to determine are $H_{0,1m}$, $H_{1,1m}$, and $H_{2,1m}$, along with the fluid variables $V_{1m}^{(1,0)}$, $W_{1m}^{(1,0)}$, and $\rho_{1m}^{(1,1)}$. Decomposing the metric into its $(1,0)$ and $(1,1)$ components, we derive the corresponding perturbation equations by applying Einstein equations together with the $t$, $r$, and $\theta$ components of $\nabla_\mu T^{\mu\nu} = 0$. The $(1,0)$ vacuum equations for $H_{0,1m}$, $H_{1,1m}$, and $H_{2,1m}$ coincide with those available in Appendix B of [80].

The functional forms of the equations for matter perturbations are identical to those in Eqs. (70)-(72), valid for modes with $\ell \geq 2$, except for the coefficients $w_1$, $w_2$, and $\gamma_1$, as well as the scattering potential $V^\rho$, whose explicit expressions are provided in the accompanying `Mathematica` file.

## 4 Gravitational wave fluxes

Having determined the axial and polar perturbations, we can compute the associated GW fluxes at infinity and at the horizon. The asymptotic structure of our metric allows us to use the standard procedure adopted in vacuum [87,88], aimed at determining the form of the perturbations in a coordinate system where the metric exhibits the correct radial dependence [89].

**In our model the matter background is a stationary, adiabatic perfect fluid distribution with no viscosity, heat conduction, or mass outflow. With these boundary conditions there is no asymptotic fluid contribution to the energy fluxes at infinity or at the horizon. The secondary does excite fluid perturbations, but their energy is stored and redistributed within the medium rather than transported across $r_\mathrm{h}$ or to $r \to \infty$. The only radiative degrees of freedom at leading order are the standard GW modes, which carry imprints of matter through coupling and background effects.**

**While no asymptotic matter fluxes are present, one may still expect local interactions between the perturber and the fluid. A Newtonian estimate suggests that a local drag force on the worldline — analogous to dynamical friction [90] — would appear at order $\mathcal{O}(q^2\epsilon)$, sharing the same radiative scaling as the flux corrections discussed in the next section. Evaluating this effect would require computing the self-consistent motion of the secondary in the perturbed geometry, i.e. feeding the metric corrections back into the worldline evolution, a SF analysis that lies beyond the scope of this work.**

To move from the RWZ gauge to the radiation gauge, we perform an infinitesimal coordinate transformation such that

$$\delta g_{\mu\nu}^{\mathrm{RG}} = \delta g_{\mu\nu}^{\mathrm{RWZ}} - \nabla_\mu \xi_\nu - \nabla_\nu \xi_\mu \,, \tag{83}$$

where $\xi^\mu$ is a gauge vector expanded in multipole components (summation over $(\ell, m)$ is implicit):

$$\xi_\mu = (\alpha_1, \alpha_2, r^2[\alpha_3 \csc\theta\, \partial_\phi + \alpha_4 \partial_\theta], r^2[\alpha_4 \partial_\phi - \alpha_3 \sin\theta\, \partial_\theta]) Y_{\ell m} \,, \tag{84}$$

with $\alpha_{1,2,3,4}$ being gauge functions dependent on $(t,r)$. Following [87], at infinity the perturbation tensor satisfies the outgoing radiation conditions:

$$\delta g_{\mu\nu}^{\mathrm{ORG}} n^\mu n^\nu = \delta g_{\mu\nu}^{\mathrm{ORG}} n^\mu m^\nu = \delta g_{\mu\nu}^{\mathrm{ORG}} n^\mu m^{\nu*} = \delta g_{\mu\nu}^{\mathrm{ORG}} n^\mu l^\nu = \delta g_{\mu\nu}^{\mathrm{ORG}} m^\mu m^{\nu*} = 0 \,, \tag{85}$$

433   where the null tetrad $(l^\mu, n^\mu, m^\mu, m^{\mu*})$ has components:

$$l_\mu = \left\{ -\sqrt{-g_{tt}g_{rr}}, g_{rr}, 0, 0 \right\},$$

$$n_\mu = -\frac{1}{2} \left\{ \sqrt{-g_{tt}/g_{rr}}, 1, 0, 0 \right\},$$

$$m_\mu = \frac{1}{\sqrt{2}} \left\{ 0, 0, r, ir\sin\theta \right\}, \tag{86}$$

434   with $l^\mu l_\mu = n^\mu n_\mu = m^\mu m_\mu = m^{\mu*}m^*_\mu = 0$, $l^\mu n_\mu = -1 = -m^\mu m^*_\mu$, and the asterisk denoting
435   complex conjugation [91].

436      Equations (85), together with the gauge transformation (83), can be used to express $\alpha_{1,2,3,4}$
437   in terms of the RWZ metric perturbations and reconstruct the perturbation tensor at infinity.[6]
438      From the asymptotic form of the polar and axial components at $r \to \infty$, using Eqs. (A.1)–(A.5),
439   we find to leading order:

$$h_{0\ell m} \simeq -h_{1\ell m} \simeq r(\phi^{(1,0)}_{\ell m} + \phi^{(1,1)}_{\ell m}), \tag{87}$$

$$H_{2,\ell m} \simeq H_{0,\ell m} \simeq -H_{1,\ell m} \simeq r\partial_t(\chi^{(1,0)}_{\ell m} + \chi^{(1,1)}_{\ell m}) - \frac{2rM_e}{\Lambda}\partial^2_t \chi^{(1,0)}_{\ell m}, \tag{88}$$

$$K_{\ell m} \simeq -(\chi^{(1,0)}_{\ell m} + \chi^{(1,1)}_{\ell m}) + \frac{2M_e}{\Lambda}\partial_t \chi^{(1,0)}_{\ell m}, \tag{89}$$

440   assuming that at spatial infinity $\partial_t = -\partial_r + \mathcal{O}(1/r)$. In the radiation zone, the perturbation
441   becomes:

$$\delta g^{\text{ORG}}_{AB} = -2r^2(\alpha_4 V^{\ell m}_{AB} + \alpha_3 W^{\ell m}_{AB}) + \mathcal{O}(1), \tag{90}$$

442   where indices $A, B$ span the angular coordinates $(\theta, \phi)$, and

$$\alpha_3 = \frac{1}{r}\int^t (\phi^{(1,0)}_{\ell m} + \phi^{(1,1)}_{\ell m})dt', \tag{91}$$

$$\alpha_4 = -\frac{1}{2r}\int^t \left( \chi^{(1,0)}_{\ell m} + \chi^{(1,1)}_{\ell m} - \frac{2M_e}{\Lambda}\partial_t \chi^{(1,0)}_{\ell m} \right)dt', \tag{92}$$

443   and

$$V_{AB} = \left( \nabla_A\nabla_B + \frac{\lambda}{2}\Omega_{AB} \right)Y_{\ell m} \quad , \quad W_{AB} = \frac{1}{2}\left[ \nabla_B\epsilon_A{}^C\nabla_C + \nabla_A\epsilon_B{}^C\nabla_C \right]Y_{\ell m}, \tag{93}$$

444   with $\Omega_{AB} = \text{diag}(1, \sin^2\theta)$, $\nabla_A$ the covariant derivative, and $\epsilon_{AB}$ the Levi-Civita tensor on the
445   unit 2-sphere.

446      The energy and angular momentum fluxes can be obtained from the Isaacson stress-energy
447   tensor for gravitational waves,

$$T^{\text{GW}}_{\mu\nu} = \frac{1}{64\pi}\langle \nabla_\mu\delta g^{\alpha\beta}\nabla_\nu\delta g_{\alpha\beta} \rangle, \tag{94}$$

448   where $\langle \ldots \rangle$ denotes average over a region of spacetime large compared with the GW wave-
449   length. Given the symmetry of the background, we can express fluxes using the Killing vectors
450   $\{\xi^\nu_{(t)}, \xi^\nu_{(\phi)}\}$ associated to the two cyclic variables $t$ and $\phi$:

$$-dE = \int_\Sigma T^{\text{GW}\mu}{}_\nu \xi^\nu_{(t)} d\Sigma_\mu = \pm\left[ \frac{|g_{tt}|}{g_{rr}} \right]^{1/2} r^2 \int_\Sigma T^{\text{GW}}_{tr} d\Omega dt, \tag{95}$$

$$dL = \int_\Sigma T^{\text{GW}\mu}{}_\nu \xi^\nu_{(\phi)} d\Sigma_\mu = \pm\left[ \frac{|g_{tt}|}{g_{rr}} \right]^{1/2} r^2 \int_\Sigma T^{\text{GW}}_{r\phi} d\Omega dt, \tag{96}$$

---

[6]These calculations are nearly identical to those in Appendix B of [87], except for the general form of the metric components $g_{tt}$ and $g_{rr}$, which include matter contributions beyond Schwarzschild.

where $d\Sigma_\mu$ is a surface element outward-oriented on $\Sigma$ and the signs $-$ and $+$ are for flux at horizon and at infinity respectively. Expanding all quantities at leading order in $1/r$, and using Eqs. (90)–(91) within the energy flux (95), to order $\mathcal{O}(q^2\epsilon)$ we obtain:

$$\dot{E}_{\ell m}^\infty = \frac{1}{64\pi} \frac{(\ell+2)!}{(\ell-2)!} \left( \left| \chi_{\ell m}^{(1,0)} \right|^2 + 4 \left| \phi_{\ell m}^{(1,0)} \right|^2 + 2 \operatorname{Re} \left[ \chi_{\ell m}^{(1,0)} \chi_{\ell m}^{(1,1)*} + 4 \phi_{\ell m}^{(1,0)} \phi_{\ell m}^{(1,1)*} \right. \right.$$
$$\left. \left. - \frac{2M_e}{\Lambda} \chi_{\ell m}^{(1,0)} \partial_t \chi_{\ell m}^{(1,0)*} \right] \right). \tag{97}$$

Similarly, for the angular momentum flux, Eq. (96), we have:

$$\dot{L}_{\ell m}^\infty = \frac{im}{128\pi} \frac{(\ell+2)!}{(\ell-2)!} \left[ \chi_{\ell m}^{(1,0)} \int^t \chi_{\ell m}^{(1,0)*} dt' + 4\phi_{\ell m}^{(1,0)} \int^t dt' \phi_{\ell m}^{(1,0)*} - \chi_{\ell m}^{(1,0)} \left( \frac{2M_e}{\Lambda} \chi_{\ell m}^{(1,0)*} \right. \right.$$
$$\left. - \int^t dt' \chi_{\ell m}^{(1,1)*} \right) - \left( \frac{2M_e}{\Lambda} \partial_t \chi_{\ell m}^{(1,0)} - \chi_{\ell m}^{(1,1)} \right) \int^t dt' \chi_{\ell m}^{*(1,0)}$$
$$\left. + 4\phi_{\ell m}^{(1,0)} \int^t dt' \phi_{\ell m}^{(1,1)*} + 4\phi_{\ell m}^{(1,1)} \int^t dt' \phi_{\ell m}^{(1,0)*} \right] + \text{c.c.} . \tag{98}$$

The first two terms in Eqs. (97) and (98) correspond to the standard fluxes at infinity for vacuum perturbations around Schwarzschild BHs.

Calculations of GW fluxes at the horizon proceed analogously to those at infinity. We impose an ingoing radiation gauge by swapping $l^\mu \leftrightarrow n^\mu$ in Eqs. (85), and express the gauge functions in terms of the RWZ metric perturbations near the horizon, i.e., in the limit $f \to 0$.

Using Eqs. (A.1)–(A.5), we obtain the leading-order behavior of the axial and polar components as $r \to 2M$:

$$f h_{1\ell m} \simeq -2M \left( \phi_{\ell m}^{(1,0)} + \phi_{\ell m}^{(1,1)} \right) + \frac{3MH_h}{2} \phi_{\ell m}^{(1,0)} , \tag{99}$$

$$h_{0\ell m} \simeq -2M \left( \phi_{\ell m}^{(1,0)} + \phi_{\ell m}^{(1,1)} \right) - \frac{M}{2} H_h \phi_{\ell m}^{(1,0)} , \tag{100}$$

$$H_{2,\ell m} \simeq H_{0,\ell m} \simeq H_{1\ell m} \simeq \frac{1}{2} (4M\partial_t - 1) \left[ \chi_{\ell m}^{(1,0)} + \chi_{\ell m}^{(1,1)} \right] - \frac{H_h}{8} (4M\partial_t - 1) \chi_{\ell m}^{(1,0)} , \tag{101}$$

$$\partial_t K_{\ell m} \simeq \left( \frac{\Lambda+1}{2M} + \partial_t \right) \left( \chi_{\ell m}^{(1,0)} + \chi_{\ell m}^{(1,1)} \right) - \frac{H_h}{4} \left( \frac{\Lambda+1}{2M} + \partial_t \right) \chi_{\ell m}^{(1,0)} , \tag{102}$$

where $H_h \equiv H(r = r_h)$, and we assume that near the horizon $\partial_t = f\partial_r + \mathcal{O}(f)$. Combining[7] these expressions with Eqs. (83) and (85), we can write the metric perturbation in the ingoing radiation gauge as:

$$\delta g_{AB}^{\text{IRG}} = -8M^2 \left( \alpha_4 V_{AB}^{\ell m} + \alpha_3 W_{AB}^{\ell m} \right) + \mathcal{O}(f) , \tag{103}$$

with the gauge coefficients given by

$$\alpha_3 = -\frac{1}{2M} \int^t \left( \phi_{\ell m}^{(1,0)} + \phi_{\ell m}^{(1,1)} - \frac{H_h}{4} \phi_{\ell m}^{(1,0)} \right) dt' , \tag{104}$$

$$\alpha_4 = -\frac{1}{4M} \int^t \left( \chi_{\ell m}^{(1,0)} + \chi_{\ell m}^{(1,1)} + \frac{MH_h}{3+2\Lambda} \partial_t \chi_{\ell m}^{(1,0)} - \frac{H_h}{4} \chi_{\ell m}^{(1,0)} \right) dt' . \tag{105}$$

---

[7]Following [87] we rescale $\alpha_2 \to \alpha_2 f^{-1}(1 - H_h)$.

467 The calculation of energy and angular momentum fluxes proceeds similarly to the far-zone
468 treatment [87], by isolating the $\mathcal{O}(f^{-1})$ contribution to the GW stress-energy tensor (94), and
469 neglecting terms of order $\mathcal{O}(1)$.

470 We substitute the expression of the metric perturbation (103) into Eqs. (95)-(96), also
471 multiplying by a $-$ sign to account that we compute BH absorption rather fluxes in the radiation
472 zone. To the leading order in $f$ we find:

$$
\dot{E}_{\ell m}^{\mathrm{H}} = \frac{1}{64\pi}\frac{(\ell+2)!}{(\ell-2)!}\left(\left|\chi_{\ell m}^{(1,0)}\right|^2 + 2\operatorname{Re}\left[\chi_{\ell m}^{(1,0)}\chi_{\ell m}^{(1,1)*} + \frac{MH_h}{3+2\Lambda}\chi_{\ell m}^{(1,0)}\partial_t\chi_{\ell m}^{(1,0)*}\right] + 4\left|\phi_{\ell m}^{(1,0)}\right|^2 \right.
$$
$$
\left. + 8\operatorname{Re}\left[\phi_{\ell m}^{(1,0)}\phi_{\ell m}^{(1,1)*}\right]\right). \tag{106}
$$

$$
\dot{L}_{\ell m}^{\mathrm{H}} = \frac{im}{128\pi}\frac{(\ell+2)!}{(\ell-2)!}\left[\chi_{\ell m}^{(1,0)}\int^t \chi_{\ell m}^{(1,0)*}dt' + 4\phi_{\ell m}^{(1,0)}\int^t dt'\phi_{\ell m}^{(1,0)*}\right.
$$
$$
+ \chi_{\ell m}^{(1,0)}\int^t dt'\left(\frac{MH_h}{3+2\Lambda}\partial_t\chi_{\ell m}^{(1,0)*} + \chi_{\ell m}^{(1,1)*}\right) + \left(\frac{MH_h}{3+2\Lambda}\partial_t\chi_{\ell m}^{(1,0)} + \chi_{\ell m}^{(1,1)}\right)\int^t dt'\chi_{\ell m}^{(1,0)*}
$$
$$
\left. + 4\phi_{\ell m}^{(1,0)}\int^t dt'\phi_{\ell m}^{(1,1)*} + 4\phi_{\ell m}^{(1,1)}\int^t dt'\phi_{\ell m}^{(1,0)*}\right] + \text{c.c.} \, . \tag{107}
$$

473 The first two terms in Eqs. (106)–(107) represent vacuum contributions to the energy and
474 angular momentum fluxes. The remaining terms depend on the matter distribution and vanish
475 in the limit $\epsilon \to 0$.

# 5 Conclusions

477 In this work, we developed a multi-parameter framework to model the dynamics and GW
478 emission of binaries with large mass asymmetries embedded in dense astrophysical environ-
479 ments. Previous studies have emphasized the scientific potential of such systems to probe the
480 properties of baryonic and dark matter evolving alongside compact objects [27,61,64]. How-
481 ever, these efforts also highlighted the significant complications introduced by non-vacuum
482 environments, which have so far made accurate waveform modeling intractable.

483 Motivated by these challenges, we constructed a semi-analytical approach that treats mat-
484 ter effects as small perturbations to vacuum spacetime, as supported by most realistic astro-
485 physical scenarios. By expanding Einstein equations around the Schwarzschild solution in
486 powers of the binary mass ratio and the ratio of environmental to BH density, we derived
487 expressions for both metric and matter perturbations within a genuinely SF framework at adi-
488 abatic order.

489 Our key results, summarized in Eqs. (51)–(52), (67)–(68), and (70)–(72), show that both
490 axial and polar perturbations reduce to equations closely resembling the well-known Regge-
491 Wheeler and Zerilli formalisms. Notably, unlike previous studies, we demonstrate that polar
492 modes can be captured by a single Zerilli-like master variable, greatly simplifying numerical
493 computations. We provide explicit expressions for reconstructing the metric functions and
494 computing GW fluxes for binaries on generic orbits.

495 This framework represents an initial step toward the development of accurate and compu-
496 tationally feasible waveform models for asymmetric binaries in complex environments — key
497 targets for future GW detectors like LISA. It also offers a flexible tool to study the interaction
498 of such systems with ambient matter via time-domain evolution, and to investigate properties
499 typically studied in vacuum, such as BH quasinormal mode spectra [92,93]. However, several
500 advancements are necessary to reach full astrophysical realism.

One major, yet essential, challenge lies in modeling binaries with a rotating primary. Describing matter perturbations around Kerr BHs could benefit from recent progress in modeling vacuum perturbations within modified gravity theories, assuming small deviations from GR [94–97]. In principle, the BH spin could be introduced as a third perturbative parameter within a slow-rotation scheme, such as the Hartle-Thorne formalism [98]. However, this approach generally exhibits poor convergence at high spin values, which are expected for astrophysical BHs. The fluid description could also be enhanced in multiple ways, for instance by investigating the impact of viscous effects on the binary dynamics [99].

**While our focus here is methodological, and the present model still has limited direct astrophysical applicability, due to the absence of spin and the restriction to spherically symmetric matter topologies, it nonetheless provides a first consistent framework for matter-embedded compact binaries. Interestingly, spherically symmetric configurations of BHs immersed in dense gas could, in fact, be relevant to certain recently observed compact sources — although at high redshift — the so-called "red dots," which may represent heavily enshrouded accreting BHs [100–102].**

Finally, current studies of the evolution of asymmetric binaries including radiation reaction have mostly been restricted to circular, equatorial orbits due to computational complexity (see Ref. [103] for a study on the relevance of eccentricity in binaries immersed in an accretion disk). The framework developed here allows exploration of EMRI and IMRI dynamics on generic, eccentric, and inclined orbits across a broad parameter space, and assessment of the impact of matter on parameter estimation using recent tools developed to analyze GW signals from asymmetric binaries [104–106].

# Acknowledgements

We thank Rodrigo Vicente, Laura Sberna and Konstantinos Kritos for interesting and useful discussions. We also thank Vitor Cardoso, Nicholas Speeney, Richard Brito, Kyriakos Destounis for having carefully read this manuscript and for useful comments. We also thank the Referees for their valuable comments, which have helped improve the quality of this manuscript.

**Funding information**     A.M. acknowledges financial support from MUR PRIN Grants No. 2022-Z9X4XS and No. 2020KB33TP. S.D. acknowledges financial support from MUR, PNRR - Missione 4 - Componente 2 - Investimento 1.2 - finanziato dall'Unione europea - NextGenerationEU (cod. id.: SOE2024_0000167, CUP:D13C25000660001).

# A    Metric perturbations as a function of the master variables

Metric perturbations can be easily reconstructed once a solution for the master equations (47)-(48) and (67)-(68) have been found. In this Appendix we provide relations that determine axial and polar metric functions at the linear order in $\mathcal{O}(\epsilon)$. In the Regge-Wheeler gauge for axial modes with $\ell \geq 2$ we have:

$$h_{1,\ell m} = -rf^{-1}\phi_{\ell m}^{(1,0)} + \frac{3[frH-2m]}{4f^2}\phi_{\ell m}^{(1,0)} - rf^{-1}\phi_{\ell m}^{(1,1)} \,, \tag{A.1}$$

$$\partial_t h_{0,\ell m} = -f\,\partial_r\left(r\phi_{\ell m}^{(1,0)}\right) + f\frac{8i\sqrt{2}\pi r^2 \mathcal{D}_{\ell m}^{(1,0)}}{\sqrt{\lambda(\lambda-2)}} + f\frac{8i\sqrt{2}\pi r^2[H\mathcal{D}_{\ell m}^{(1,0)} + \mathcal{D}_{\ell m}^{(1,1)}]}{\sqrt{\lambda(\lambda-2)}}$$

$$-f\,\partial_r\left(r\phi_{\ell m}^{(1,1)}\right) + \frac{1}{4}\left(\frac{2m}{r} - fH\right)\partial_r\left(r\phi_{\ell m}^{(1,0)}\right) + \left[\frac{m}{fr} + 2\pi r^2(p_r - \rho)\right]\phi_{\ell m}^{(1,0)} \tag{A.2}$$

where $f = 1 - 2M/r$ and $\lambda = \ell(\ell + 1)$. Frequency domain expressions can be obtained by replacing time derivatives as $\partial_t \to -i\omega$.

The reconstruction of polar perturbations is more convoluted. We provide here explicit expressions including only the master functions. The full form depending on the coefficients of the secondary stress-energy tensor is provided in the `Mathematica` supplementary file:

$$\partial_t H_{0,\ell m} = [A_1 + A_5 + (A_2 + A_6)\partial_r + A_3\partial_r^2 + A_4\partial_r^3]\chi_{\ell m}^{(1,0)} + (A_1 + A_2\partial_r + rf\partial_r^2)\chi_{\ell m}^{(1,1)}$$
$$+ (A_7 + B_4\partial_r)V_{\ell m}^{(1,0)} + (A_8 + B_5\partial_r)W_{\ell m}^{(1,0)} + S_{\ell m}^{H_0}\,, \tag{A.3}$$

$$H_{1,\ell m} = (B_1 + B_6 + B_2\partial_r + B_3\partial_r^2)\chi_{\ell m}^{(1,0)} + (B_1 + r\partial_r)\chi_{\ell m}^{(1,1)} + \frac{B_4}{f}V_{\ell m}^{(1,0)} + \frac{B_5}{f}W_{\ell m}^{(1,0)} + S_{\ell m}^{H_1}\,, \tag{A.4}$$

$$\partial_t K_{\ell m} = \left(C_1 + C_4 + \frac{f}{r}B_2\partial_r + \frac{f}{r}B_3\partial_r^2\right)\chi_{\ell m}^{(1,0)} + (C_1 + f\partial_r)\chi_{\ell m}^{(1,1)} + \frac{B_4}{r}V_{\ell m}^{(1,0)} + \frac{B_5}{r}W_{\ell m}^{(1,0)} + S_{\ell m}^{K}\,, \tag{A.5}$$

542  where the source terms $S_{\ell m}^{H_0,H_1,K}$ depend on the particle orbital motion, and

$$A_1 = -\frac{9M^3 + 9M^2 r\Lambda + 3Mr^2\Lambda^2 + r^3\Lambda^2(1+\Lambda)}{r^2\mathcal{C}^2} \quad , \quad A_2 = \frac{3M^2 - Mr\Lambda + r^2\Lambda}{r\mathcal{C}} \,, \tag{A.6}$$

$$A_3 = rf - \frac{rfH}{4} - \frac{4\pi f r^4(2p_r - 3\rho)}{\mathcal{C}} + \frac{m[(2-3\Lambda)r - 13M]}{2\mathcal{C}} \quad , \quad A_4 = -\frac{2r^2 f m}{\mathcal{C}} \,, \tag{A.7}$$

$$\begin{aligned}
A_5 = &-\frac{A_1 H}{4} + \frac{m}{2r^4 f^2 \mathcal{C}^4}\Big\{9(14\Lambda - 3)M^5 r - 54M^6 + 3[\Lambda(124\Lambda - 33) + 36]M^4 r^2 \\
&+ 60(\Lambda - 1)\Lambda(2\Lambda - 3)M^3 r^3 + 3\Lambda^2[2\Lambda(7\Lambda - 6) + 55]M^2 r^4 + \Lambda^3[\Lambda(2\Lambda - 33) - 6]Mr^5 \\
&+ \Lambda^3[12 - (\Lambda - 9)\Lambda]r^6\Big\} - \frac{2\pi\rho}{f\mathcal{C}^3}\Big\{18M^4 + 9(1 - 4\Lambda)M^3 r + 6\Lambda(\Lambda + 12)M^2 r^2 \\
&+ \Lambda[(3 - 4\Lambda)\Lambda - 12]Mr^3 + 4\Lambda^2(\Lambda + 1)r^4\Big\} - \frac{4\pi r}{\mathcal{C}^2}\big[15M^2 + 6\Lambda Mr + \Lambda(3\Lambda + 4)r^2\big]p_t \\
&+ \frac{2\pi r^2}{\mathcal{C}^2}\big[3M^2 + \Lambda(\Lambda + 2)r^2\big]\rho' - \frac{2\pi}{f\mathcal{C}^3}\Big\{72M^4 + 3(34\Lambda - 15)M^3 r + 6\Lambda(2\Lambda - 15)M^2 r^2 \\
&+ \Lambda[\Lambda(6\Lambda - 5) + 12]Mr^3 - 4\Lambda^2(\Lambda + 1)r^4\Big\}p_r \,, \tag{A.8}
\end{aligned}$$

$$\begin{aligned}
A_6 = &-\frac{A_2 H}{4} + \frac{4\pi r^2 \rho}{\mathcal{C}^2}\big[(6 - 9\Lambda)Mr + 4\Lambda r^2 - 15M^2\big] + \frac{4\pi r^2}{\mathcal{C}}[M - (\Lambda + 2)r]p_r \\
&- \frac{16\pi r^3 f p_t}{\mathcal{C}} + \frac{4\pi r^4 f \rho'}{\mathcal{C}} - \frac{m}{2r^2 f\mathcal{C}^3}\Big\{9M^4 + (36 - 69\Lambda)M^3 r - 9(\Lambda - 13)\Lambda M^2 r^2 \\
&+ \Lambda[(14 - 11\Lambda)\Lambda - 12]Mr^3 + \Lambda^2(9\Lambda + 8)r^4\Big\} \,, \tag{A.9}
\end{aligned}$$

$$A_7 = \frac{4f}{\mathcal{C}^2}[3M^2 + r^2\Lambda^2 + Mr(2\Lambda - 3)]\kappa_t - \frac{4r^2 f^2}{\mathcal{C}}\kappa_t' \,, \tag{A.10}$$

$$A_8 = \frac{2rf}{\mathcal{C}^2}\Big\{rf\mathcal{C}(r\rho' - 2p_t) - [9M^2 + (5M - r)r\Lambda]p_r + [3M(2r - M) + r(M + r)\Lambda]\rho\Big\} \,, \tag{A.11}$$

$$B_1 = \frac{\Lambda r}{\mathcal{C}} - \frac{M}{fr} \quad , \quad B_2 = r - \frac{rH}{4} - \frac{m}{2f\mathcal{C}^2}[3M^2 + 6Mr(1 + \Lambda) - r^2\Lambda(2 + \Lambda)] \,, \tag{A.12}$$

$$+ \frac{4\pi r^4(\rho - 2p_r)}{\mathcal{C}} \quad , \quad B_3 = \frac{2r^2 m}{\mathcal{C}} \quad , \quad B_4 = -\frac{4r^2 f^2 \kappa_t}{\mathcal{C}} \,, \tag{A.13}$$

$$\begin{aligned}
B_5 = &-\frac{2f^2 r^3 \kappa_r}{\mathcal{C}} \quad , \quad B_6 = \frac{2\pi r^2}{f\mathcal{C}^2}\big[3M^2 + \Lambda(\Lambda + 2)r^2\big]\rho - \frac{H}{4}B_1 \\
&- \frac{3m}{2r^2 f^2 \mathcal{C}^3}\Big[33M^4 + (31\Lambda + 6)M^3 r + \Lambda^2(\Lambda + 2)r^4 + 3\Lambda(5\Lambda + 3)M^2 r^2 \\
&+ \Lambda(\Lambda^2 + 2)Mr^3\Big] - \frac{2\pi r^2}{f\mathcal{C}^2}\big[15M^2 + 6\Lambda Mr + \Lambda(3\Lambda + 4)r^2\big]p_r \,, \tag{A.14}
\end{aligned}$$

$$\begin{aligned}
C_1 = &\ \frac{6M^2 + 3Mr\Lambda + r^2\Lambda(1 + \Lambda)}{r^2\mathcal{C}} \quad , \quad C_4 = -\frac{H}{4}C_1 + \frac{2\pi r}{\mathcal{C}^2}\rho\big[3M^2 + \Lambda(\Lambda + 2)r^2\big] \\
&- \frac{m}{2fr^3\mathcal{C}^3}\Big[18M^4 - 3(5\Lambda - 6)M^3 r - 9(\Lambda - 3)\Lambda M^2 r^2 - 3\Lambda(3\Lambda^2 - 2)Mr^3 \\
&- \Lambda^2((\Lambda - 3)\Lambda - 6)r^4\Big] - \frac{2\pi r}{\mathcal{C}^2}\big[15M^2 + 6\Lambda Mr + \Lambda(3\Lambda + 4)r^2\big]p_r \,, \tag{A.15}
\end{aligned}$$

543  with $\Lambda = (\ell + 2)(\ell - 1)/2$, $\kappa_{t,r} = p_{t,r} + \rho$, $\mathcal{C} = r\Lambda + 3M$ and a prime denoting radial derivative.

## B Decoupling of Axial and Polar Modes into vacuum and matter components using the scaling function $Z$.

In this appendix, we clarify why, in computing axial and polar modes, we chose to work with a single metric perturbation rather than separating vacuum $(0,1)$ and matter $(1,1)$ components from the beginning.

The structure of the equations for axial modes, for example, allows one to follow a procedure similar to the vacuum case. In this framework, it is possible to eliminate one of the metric functions at each order in $\epsilon$ using the Einstein equations, leading to two second-order differential equations in $(r,t)$ for the $(1,0)$ and $(1,1)$ perturbations. These equations can then be recast in the familiar wave-like form by introducing a generalized tortoise coordinate, which facilitates the imposition of boundary conditions at spatial infinity and the BH horizon. However, a subtlety arises from the fact that the generalized tortoise coordinate $dr^\star/dr = 1/\sqrt{-g_{tt}/g_{rr}}$, depends on the parameter $\epsilon$. This introduces an ambiguity due to the perturbative relation between $r$ and $r_\star$, since $dr^\star/dr = f^{-1} + \mathcal{O}(\epsilon)$, on whether one should use $r$ or $r_\star$ in the perturbative expansion (see [107] for further details). This issue can be circumvented by following the approach developed in [108], which we briefly outline here.

Consider a scalar perturbation $\Phi$ on a fixed, spherically symmetric background with the metric:

$$ds^2 = -A(r)dt^2 + \frac{dr^2}{B(r)} + r^2(d\theta^2 + \sin^2\theta\, d\phi^2). \tag{B.1}$$

After decomposing $\Phi$ into spherical harmonics, the Klein–Gordon equation $\Box\Phi = 0$ can be written as:

$$-\frac{\partial^2\Phi}{\partial t^2} + \mathcal{F}\frac{d}{dr}\left(\mathcal{F}\frac{d\Phi}{dr}\right) - \mathcal{F}V\Phi = 0, \tag{B.2}$$

where $V$ is the effective potential, which depends on the background geometry. Assume the metric functions $A(r)$ and $B(r)$ are close to the Schwarzschild solution:

$$A(r) = \left(1 - \frac{r_h}{r}\right)(1 + \delta A), \quad B(r) = \left(1 - \frac{r_h}{r}\right)(1 + \delta B), \tag{B.3}$$

with $\delta A, \delta B \ll 1$, and where $r_h$ denotes the horizon radius.[8] Then, at the leading order in the metric changes $(\delta A, \delta B)$, the function $\mathcal{F} = \sqrt{AB}$ can be expressed as:

$$\mathcal{F} = f(r)Z(r) = \left(1 - \frac{r_h}{r}\right)Z(r) = \left(1 - \frac{r_h}{r}\right)[1 + \delta Z(r)].$$

Introducing the rescaled field $\phi = \sqrt{Z}\Phi$, and expanding Eq. (B.2) to linear order in $\delta Z$, the master equation becomes:

$$-(1 + 2\delta Z)\frac{\partial^2\phi}{\partial t^2} + f\frac{d}{dr}\left(f\frac{d\phi}{dr}\right) - f\tilde{V}\phi = 0, \tag{B.4}$$

where $\tilde{V}$ is the modified potential (the explicit form can be found in [108]). For both axial and polar sectors, the metric perturbations we find satisfy master equations analogous to Eq. (B.2), with $r_h = 2M$, and can be recast into the form of Eq. (B.4) by introducing an appropriate scaling function $Z$. Since the prefactor of the radial derivative terms in Eq. (B.4) is $f(r)$, we can adopt the standard tortoise coordinate $r^\star = r + 2M\ln(r/2M - 1)$. This allows us to write the perturbations as a sum of the $(1,0)$ and $(1,1)$ components, and isolate their contributions without introducing ambiguities.

---

[8]Note that in general $r_h$ may differ from the Schwarzschild value. In such cases, $r_h$ should be treated as a fundamental parameter in the computation of perturbations, as done in the cases studied in [108].

## C  Coefficients of the particle stress-energy momentum tensor

The form of the coefficients $\{\mathcal{A}_{\ell m}^{0(1,0)}, \ldots \mathcal{F}_{\ell m}^{(1,0)}\}$ of the particle stress-energy tensor, can be found by projecting each one of the ten tensor harmonics on Eq. (30). Introducing the scalar product between two tensor harmonics $A_{\mu\nu}$ and $B_{\mu\nu}$:

$$(A, B) = \int\int \eta^{\mu\sigma}\eta^{\nu\delta}A^*\mu\nu B_{\sigma\delta}\sin\theta d\theta d\phi \,, \tag{C.1}$$

where $\eta_{\mu\nu}$ is the Minkowski metric tensor in spherical coordinates, and $*$ denotes complex conjugation, we have, for example, $\mathcal{A}_{\ell m}^{(1,1)} = (\mathbf{a}_{\ell m}, T^{p(1,1)})$. We provide the expression of the coefficients for generic orbits in the supplementary material. In the case of equatorial circular motion, $\theta_p = \pi/2$, for a secondary at a radius $r = r_p$, the only non vanishing coefficients are given by $Q_{\ell m}^{0(1,0)}$ for the axial sector, and $(\mathcal{A}_{\ell m}^{0(1,0)}, \mathcal{B}_{\ell m}^{0(1,0)}, \mathcal{G}_{\ell m}^{0(1,0)}, \mathcal{D}_{\ell m}^{0(1,0)}, \mathcal{F}_{\ell m}^{0(1,0)})$ for the polar modes (and similarly for the $(1, 1)$ coefficients). Their explicit form is given by:

$$\mathcal{Q}^{0(1,0)} = \frac{\sqrt{2}f\mathcal{L}^{(0,0)}}{r^3\sqrt{\lambda}}\partial_\theta Y_{\ell m}^*\delta(r - r_p) \,, \quad \mathcal{A}^{0(1,0)} = \frac{f\mathcal{E}^{(0,0)}}{r^2}Y_{\ell m}^*\delta(r - r_p) \,, \tag{C.2}$$

$$\mathcal{B}^{0(1,0)} = \frac{i\sqrt{2}f\mathcal{L}^{(0,0)}}{r^3\sqrt{\lambda}}\partial_\phi Y_{\ell m}^*\delta(r - r_p) \,, \tag{C.3}$$

$$\mathcal{D}^{(1,0)} = \frac{i\sqrt{2}f(\mathcal{L}^{(0,0)})^2}{r^4\mathcal{E}^{(0,0)}\sqrt{\lambda(\lambda - 2)}}\partial_{\theta\phi}Y_{\ell m}^*\delta(r - r_p) \,, \tag{C.4}$$

$$\mathcal{F}^{(1,0)} = \frac{f(\mathcal{L}^{(0,0)})^2\delta(r - r_p)}{\sqrt{2}r^4\mathcal{E}^{(0,0)}\sqrt{\lambda(\lambda - 2)}}[\partial_\phi^2 - \partial_\theta^2]Y_{\ell m}^* \quad , \quad \mathcal{G}^{(1,0)} = \frac{f(\mathcal{L}^{(0,0)})^2}{\sqrt{2}r^4\mathcal{E}^{(0,0)}}Y_{\ell m}^*\delta(r - r_p) \,, \tag{C.5}$$

$$Q_{\ell m}^{0(1,1)} = \frac{1}{\sqrt{2\lambda}r^4}[2fr\mathcal{L}^{(0,1)} + fr\mathcal{L}^{(0,0)}H - 2\mathcal{L}^{(0,0)}m]\partial_\theta Y_{\ell m}^*\delta(r - r_p) \,, \tag{C.6}$$

$$\mathcal{A}_{\ell m}^{0(1,1)} = \frac{1}{2r^3}[2fr\mathcal{E}^{(0,1)} + (rfH - 2m)\mathcal{E}^{(0,0)}]Y_{\ell m}^*\delta(r - r_p) \,, \tag{C.7}$$

$$\mathcal{B}_{\ell m}^{0(1,1)} = \frac{i}{\sqrt{2\lambda}r^4}[2fr\mathcal{L}^{(0,1)} + fr\mathcal{L}^{(0,0)}H - 2\mathcal{L}^{(0,0)}m]\partial_\phi Y_{\ell m}^*\delta(r - r_p) \,, \tag{C.8}$$

$$\begin{aligned}\mathcal{D}_{\ell m}^{0(1,1)} = &\frac{i\mathcal{L}^{(0,0)}\delta(r - r_p)}{r^5(\mathcal{E}^{(0,0)})^2\sqrt{2\lambda(\lambda - 2)}}\{4fr\mathcal{E}^{(0,0)}\mathcal{L}^{(0,1)} - 2fr\mathcal{E}^{(0,1)}\mathcal{L}^{(0,0)} \\ &+ \mathcal{E}^{(0,0)}\mathcal{L}^{(0,0)}[frH - 2m]\}\partial_{\theta\phi}Y_{\ell m}^* \,,\end{aligned} \tag{C.9}$$

$$\begin{aligned}\mathcal{F}_{\ell m}^{(1,1)} = &\frac{\mathcal{L}^{(0,0)}\delta(r - r_p)}{r^5(\mathcal{E}^{(0,0)})^2\sqrt{8\lambda(\lambda - 2)}}\{4fr\mathcal{E}^{(0,0)}\mathcal{L}^{(0,1)} - 2fr\mathcal{E}^{(0,1)}\mathcal{L}^{(0,0)} \\ &+ \mathcal{E}^{(0,0)}\mathcal{L}^{(0,0)}[frH - 2m]\}[\partial_\phi^2 - \partial_\theta^2]Y_{\ell m}^* \,,\end{aligned} \tag{C.10}$$

$$\mathcal{G}_{\ell m}^{(1,1)} = \frac{\mathcal{L}^{(0,0)}\delta(r - r_p)}{2\sqrt{2}r^5(\mathcal{E}^{(0,0)})^2}\{4fr\mathcal{E}^{(0,0)}\mathcal{L}^{(0,1)} - 2fr\mathcal{E}^{(0,1)}\mathcal{L}^{(0,0)} + \mathcal{E}^{(0,0)}\mathcal{L}^{(0,0)}[frH - 2m]\}Y_{\ell m}^* \,, \tag{C.11}$$

where spherical harmonics are evaluated at $\theta = \theta_p(t)$ and $\phi = \phi_p(t)$, while $\mathcal{E}^{(0,1)}$ and $\mathcal{L}^{(0,1)}$ are the non-vacuum corrections to the particle energy and angular momentum given by the $\mathcal{O}(\epsilon)$ terms in Eqs. (20)-(21).

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
