# Peer review of "A multi-parameter expansion for the evolution of asymmetric binaries in astrophysical environments"

_SciPost Physics_

## Round 1 · Referee Report · Dongjun Li (Referee 1) · 2025-8-28

Strengths

  1. This work provides a novel pathway for modeling EMRIs under a large subset of environmental effects that admits a fluid description. This is an urgent need in the waveform modeling community, as emphasized by multiple LISA working groups.
  2. The procedures in this work are concrete and potentially implementable once an equation of state of the fluid is chosen.
  3. The presentation is clear and intelligible without ambiguities.

Report

This work develops a framework for modeling extreme mass-ratio inspirals (EMRIs) embedded in generic, low-density, fluid-like environments using a multi-parameter expansion. Focusing on the case of a non-rotating central black hole, the authors expand the metric and fluid parameters (e.g., density and velocity) in terms of the mass ratio $q$ and the density ratio $\epsilon$ between the fluid and the black hole. They then systematically derive the equations of motion for gravitational and fluid perturbations up to $\mathcal{O}(q^1\epsilon^1)$. At $\mathcal{O}(q^1\epsilon^0)$ and $\mathcal{O}(q^1\epsilon^1)$, gravitational perturbations are decomposed into axial and polar sectors, following the Regge–Wheeler and Zerilli-Moncrief formalism. In each sector, the authors construct modified master variables that reduce the dynamics to a single equation for the gravitational perturbation. In the polar sector, the perturbed fluid density enters the gravitational equation at $\mathcal{O}(q^1\epsilon^1)$. Since the fluid perturbation itself is sourced only by the gravitational perturbation at $\mathcal{O}(q^1\epsilon^0)$, this system of equations can be solved hierarchically. In the end, the authors derive the modified infinity and horizon fluxes in terms of these perturbed gravitational and fluid master variables.

Overall, this work lays down a novel and systematic framework for studying EMRIs in a generic fluid environment, which is an important but long-standing challenge for accurate waveform modeling. It extends previous studies, such as [61–64], which were restricted to spherically symmetric matter distributions. While the present analysis is limited to circular orbits around non-rotating black holes, the framework appears well-positioned for generalization to rotating primaries and more generic orbits, given much recent progress in modeling ringdown in modified gravity and EMRIs embedded in scalar clouds for rotating primaries. The procedures provided by this work are concrete, and the presentation is clear. Therefore, I am confident that this work meets all the acceptance criteria of SciPost. This work could be published after the following minor suggestions are addressed.

Requested changes

  1. In Eq. (15), the perturbed metric due to the fluid also has nonzero $h_{\theta\theta}^{(0,1)}=r^2$ and $g_{\phi\phi}^{(0,1)}=r^2\sin^2{\theta}$ components, but without any additional functions multiplying them. Is this a typo (i.e., the perturbations are actually only on $tt$ and $rr$ components)?
  2. Is the equation of state relating the pressure and density in Eq. (36) valid at all orders? For example, shall I expect a different equation of state at order (0,1)? Also, could the pressure at order (1,1) also depend on the density perturbations at order (0,1)? More explanations of this part might be helpful.
  3. Could the authors comment on why the axial gravitational perturbations do not couple to the fluid parameter (except its velocity at order (1,0)), but the polar perturbations do?

Recommendation

Publish (surpasses expectations and criteria for this Journal; among top 10%)

  • validity: top
  • significance: top
  • originality: top
  • clarity: top
  • formatting: perfect
  • grammar: perfect

Author:  Sayak Datta  on 2025-11-12  [id 6023]

(in reply to Report 1 by Dongjun Li on 2025-08-28)
Disclosure of Generative AI use

The comment author discloses that the following generative AI tools have been used in the preparation of this comment:

Chatgpt

Category:
answer to question

Overall, this work lays down a novel and systematic framework for studying EMRIs in a generic fluid environment, which is an important but long-standing challenge for accurate waveform modeling. It extends previous studies, such as [61–64], which were restricted to spherically symmetric matter distributions. While the present analysis is limited to circular orbits around non-rotating black holes, the framework appears well-positioned for generalization to rotating primaries and more generic orbits, given much recent progress in modeling ringdown in modified gravity and EMRIs embedded in scalar clouds for rotating primaries.

Requested changes:

Comment 1.
In Eq. (15), the perturbed metric due to the fluid also has nonzero $h^{(0,1)}(\theta,\theta) = r^2$ and $h^{(0,1)}(\theta,\phi) = r^2 \sin^2\theta$ components, but without any additional functions multiplying them. Is this a typo (i.e., the perturbations are actually only on $tt$ and $rr$ components)?

response: We thank the referee for pointing it out. Yes, this is a typo and we have corrected it now. Now it is Eq. 17.

Comment 2.
Is the equation of state relating the pressure and density in Eq. (36) valid at all orders? For example, shall I expect a different equation of state at order (0,1)? Also, could the pressure at order (1,1) also depend on the density perturbations at order (0,1)? More explanations of this part might be helpful.

response: Since the physical properties of matter should not change under the point-particle perturbation, the underlying equation of state (EoS) is expected to remain the same. This is analogous to what is usually assumed when perturbing compact stars, where one adopts a barotropic and adiabatic approximation.

Throughout the paper we assume a barotropic equation of state without specifying its explicit form, as it is not required for the analytical developments (although it will be necessary for numerical evaluation). Equation (36) introduces the sound speeds in terms of linear perturbations, as is standard in fluid dynamics, where they appear as background fields — scalar functions of r. At higher orders (e.g., the quadratic (2,1) terms), one could relate pressure and density by including a compressibility parameter.

In our formulation, the first-order mixed perturbations (1,1) of density and pressure are proportional to each other and depend on the background quantities through the master equation (72) (of the new version), in particular via its source term, which contains both background and vacuum-perturbation contributions (1,0) in our notation. Solving this master equation yields $\rho^{(1,1)}$, and $p^{(1,1)}$ then follows directly from Eq. (36). We have added a clarifying comment in the main text after the definition of the master equation for $\rho^{(1,1)}$.

Comment 3.
Could the authors comment on why the axial gravitational perturbations do not couple to the fluid parameter (except its velocity at order (1,0)), but the polar perturbations do?

response: Axial perturbations do not couple, at this order, to the energy density or pressure perturbations because of parity considerations. These quantities are scalars and therefore transform as even-parity variables. The only fluid variable that couples to the tensorial (axial) sector is the axial component of the fluid velocity. This decoupling appears in the
non-rotating case only. When rotation is included, we would expect that axial and polar perturbations can mix; in that case, fluid motion can couple to axial perturbations. We have added a comment in the main text to clarify this point.

As a final remark, we emphasize that our formalism is not limited to circular orbits. The coefficients of the stress–energy tensor listed in the Appendix are shown for circular trajectories purely for clarity, but the framework itself accommodates generic orbital configurations. The accompanying Mathematica notebook includes all the necessary expressions to evaluate these coefficients for arbitrary orbits.

---

## Round 1 · Referee Report · Anonymous (Referee 2) · 2025-9-2

Strengths

-Sets up and completes a necessary and difficult technical computation
-Pushes forward the field in the computation of EMRI dynamics in the presence of matter fields

Weaknesses

-Paper can be quite hard to parse at times - Does not acknowledge many of its limitations - The regime of astrophysical applicability is sparsely addressed

Report

———————————————————————————————————————————————————————————————————
In this manuscript, the authors present a prescription for calculating gravitational-wave fluxes for a small body in the presence of a Schwarzschild black hole and a surrounding perfect fluid. They employ a two-parameter expansion framework to perturbatively calculate the gravitational fluxes.

This manuscript is of very high quality and will undoubtedly be of considerable use to the community. However, I do believe there are several points where the manuscript should be improved before I can recommend it for publication.

———————————————————————————————————————————————————————————————————
Section 1
———————————————————————————————————————————————————————————————————
In this section, the authors set up their equations using a point-particle prescription. In the vacuum case, this system of equations can be derived by noting that, in the global spacetime, the full metric admits an expansion of the form g = g^{Sch} + q h. Near the smaller object, this prescription becomes invalid, and the full metric instead admits an expansion of the form g = g^{secondary} + q H, where h and H represent distinct perturbations. By performing a matching procedure between these two regimes of validity, one arrives at the point-particle prescription (see https://arxiv.org/pdf/1703.02836).

In principle, a similar prescription should be followed for the fluid. The fluid near the particle may, in many cases, not be perturbatively close to the background flow near the small body—particularly within the object’s Hill sphere. As a first study, neglecting this fact may be justified; however, ignoring it may result in missing contributions to the equations of motion. These contributions may enter at higher order in the dynamics, but at present, it is not clear how they will manifest. In any case, I believe the authors should comment on this issue.
———————————————————————————————————————————————————————————————————
General Note on Sections 2 & 3
———————————————————————————————————————————————————————————————————
Before addressing specific comments, I wish to make a general observation. These sections contain a number of quite involved and impressive computations, resulting in lengthy expressions decomposed into spherical harmonics and expressed in terms of newly defined coefficients such as {H, W, V, U}. I fear, however, that many readers will find these sections difficult to parse. I suggest the authors consider whether the readability of these sections can be improved. One possibility would be to explicitly provide the covariant form of the linearised equations at each order prior to presenting their decomposed form. I leave it to the authors to decide whether they wish to take this suggestion on board.
———————————————————————————————————————————————————————————————————
Section 2
———————————————————————————————————————————————————————————————————
In this section, the authors carry out the two-parameter expansion. On at least two occasions (specifically, in equations (12) and (23)), the authors introduce undefined operators acting on the background metric and the metric perturbations. The authors should make explicit what these operators represent—i.e., whether they correspond to the second and third variations of the Einstein operator, respectively. This should be stated and expressed explicitly in the manuscript.

In addition, it would be helpful if the authors were to isolate the linearised equations at each order. As the manuscript currently stands, it is not clear whether all relevant terms are being included at each order. For example, at order (1,1), one would expect the equation to take the form

G^{(1)}_{\mu\nu}[g^{(1,1)}] = T_e^{(1,1)} + T_p^{(1,1)} - G^{(2)}[g^{(1,0)}, g^{(0,1)}],

where

G^{(n )}_{\mu\nu}[h] = (1/n!) d^n/dλ^n [ G_{\mu\nu}[ g + λ h ]] |_{λ = 0}.

In its current form, it is not clear whether the G^{(2)} terms are included in the source for g^{(1,1)}.
———————————————————————————————————————————————————————————————————
Section 3
———————————————————————————————————————————————————————————————————
In this section, the authors present their fully linearised equations. However, in the case of a perfect fluid, there exist specific regimes in which such a linearisation is valid, and others in which it cannot generally be performed. For example, in the thin-disk limit, one should only consider the linearised fluid equations when the thermal mass of the secondary is much larger than the mass of the secondary itself (see https://arxiv.org/pdf/astro-ph/0010576). This would introduce an additional scale to the validity of the equations, even within a single specific setup. I believe the authors do not appropriately discuss or adequately motivate the regimes in which their equations are applicable.
———————————————————————————————————————————————————————————————————
Section 4
———————————————————————————————————————————————————————————————————
In this section, the authors derive formulas for calculating the energy and angular momentum fluxes from gravitational waves at infinity and through the horizon. However, it is my impression that the authors are missing a significant contribution at this order: namely, the energy and angular momentum fluxes through the perturbed matter field.

My understanding is that these contributions will not be encoded in the gravitational waves alone. For example, in equations (93) and (94), the authors should include an additional term in the stress-energy tensor: T = T^{GW} + T^{Fluid},

where T^{Fluid} consists of the stress-energy of the background fluid plus its perturbations. If one were to calculate the local dissipative force due to g^{(1,0)} + g^{(1,1)}, I would expect the balance law one would derive to relate the local force to the fluxes would require the contributions from both of these stress-energy contributions. In fact, it is possible that the term arising from the fluxes in T^{Fluid} may even dominate over T^{GW} at order (1,1).

———————————————————————————————————————————————————————————————————
Conclusion
———————————————————————————————————————————————————————————————————
It is my impression that this is a strong and valuable work. However, I believe the above comments must be addressed robustly, both in a revised manuscript and in the authors’ response, before I can recommend the paper for publication. In addition, while I find the results to be of publishable merit and quality, I believe the authors do not clearly explain the regimes in which their approach is valid and in particular, the shortcomings of the work, i.e. what further work would be required in order to obtain a well-founded set-up wherein one could properly evolve the EMRI dynamics.

Requested changes

1) Add a comment on how they assume the validity of the point particle prescription in the full spacetime and the potential consequences of this assumption.

2) Explicitly include the meaning of the linearised operators in equations 12 and 23

3) Add a comment on the validity of their setup from an astrophysics point of view (What can their setup actually describe?)

4) Address the concern regarding the exclusion of the matter fluxes in their computation. If the authors believe this inclusion is not needed, please justify why.

5) In the conclusion, provide a description of the advances due to this work, as well as the limitations of this work, and discuss the road ahead to how this could actually be used for EMRI evolutions.

Optional change) Improve readability and presentation of sections 2 & 3.

Recommendation

Ask for minor revision

  • validity: good
  • significance: top
  • originality: high
  • clarity: ok
  • formatting: good
  • grammar: excellent

Author:  Sayak Datta  on 2025-11-12  [id 6024]

(in reply to Report 2 on 2025-09-02)
Disclosure of Generative AI use

The comment author discloses that the following generative AI tools have been used in the preparation of this comment:

Chatgpt

Category:
answer to question

——————————————————————————————————————
Section 1
——————————————————————————————————————

In this section, the authors set up their equations using a point-particle prescription. In the vacuum case, this system of equations can be derived by noting that, in the global spacetime, the full metric admits an expansion of the form $g = g^{Sch} + q h$. Near the smaller object, this prescription becomes invalid, and the full metric instead admits an expansion of the form $g = g^{secondary} + q H$, where h and H represent distinct perturbations. By performing a matching procedure between these two regimes of validity, one arrives at the point-particle prescription (see https://arxiv.org/pdf/1703.02836).

In principle, a similar prescription should be followed for the fluid. The fluid near the particle may, in many cases, not be perturbatively close to the background flow near the small body—particularly within the object’s Hill sphere. As a first study, neglecting this fact may be justified; however, ignoring it may result in missing contributions to the equations of motion. These contributions may enter at higher order in the dynamics, but at present, it is not clear how they will manifest. In any case, I believe the authors should comment on this issue.
——————————————————————————————————————
General Note on Sections 2 and 3
——————————————————————————————————————

Before addressing specific comments, I wish to make a general observation. These sections contain a number of quite involved and impressive computations, resulting in lengthy expressions decomposed into spherical harmonics and expressed in terms of newly defined coefficients such as {H, W, V, U}. I fear, however, that many readers will find these sections difficult to parse. I suggest the authors consider whether the readability of these sections can be improved. One possibility would be to explicitly provide the covariant form of the linearised equations at each order prior to presenting their decomposed form. I leave it to the authors to decide whether they wish to take this suggestion on board.
——————————————————————————————————————
Section 2
——————————————————————————————————————

In this section, the authors carry out the two-parameter expansion. On at least two occasions (specifically, in equations (12) and (23)), the authors introduce undefined operators acting on the background metric and the metric perturbations. The authors should make explicit what these operators represent—i.e., whether they correspond to the second and third variations of the Einstein operator, respectively. This should be stated and expressed explicitly in the manuscript.

In addition, it would be helpful if the authors were to isolate the linearised equations at each order. As the manuscript currently stands, it is not clear whether all relevant terms are being included at each order. For example, at order (1,1), one would expect the equation to take the form

\begin{equation}
G^{(1)}_{\mu\nu}[g^{(1,1)}] = T_e^{(1,1)} + T_p^{(1,1)} - G^{(2)}[g^{(1,0)}, g^{(0,1)}],
\end{equation}

where

\begin{equation}
G^{(n)}_{\mu\nu}[h] = (1/n!) d^n/d\lambda^n [G_{\mu\nu}[ g + \lambda h ]] |_{\lambda = 0}.
\end{equation}

In its current form, it is not clear whether the $G^{(2)}$ terms are included in the source for $g^{(1,1)}$.
——————————————————————————————————————
Section 3
——————————————————————————————————————

In this section, the authors present their fully linearised equations. However, in the case of a perfect fluid, there exist specific regimes in which such a linearisation is valid, and others in which it cannot generally be performed. For example, in the thin disk limit, one should only consider the linearised fluid equations when the thermal mass of the secondary is much larger than the mass of the secondary itself (see https://arxiv.org/pdf/astro-ph/0010576). This would introduce an additional scale to the validity of the equations, even within a single specific setup. I believe the authors do not appropriately discuss or adequately motivate the regimes in which their equations are applicable.
——————————————————————————————————————
Section 4
——————————————————————————————————————

In this section, the authors derive formulas for calculating the energy and angular momentum fluxes from gravitational waves at infinity and through the horizon. However, it is my impression that the authors are missing a significant contribution at this order: namely, the energy and angular momentum fluxes through the perturbed matter field.

My understanding is that these contributions will not be encoded in the gravitational waves alone. For example, in equations (93) and (94), the authors should include an additional term in the stress-energy tensor: $T = T^{GW} + T^{Fluid}$, where $T^{Fluid}$ consists of the stress-energy of the background fluid plus its perturbations. If one were to calculate the local dissipative force due to $g^{(1,0)} + g^{(1,1)}$, I would expect the balance law one would derive to relate the local force to the fluxes would require the contributions from both of these stress-energy contributions. In fact, it is possible that the term arising from the fluxes in $T^{Fluid}$ may even dominate over $T^{GW}$ at order (1,1).
——————————————————————————————————————
Conclusion
——————————————————————————————————————

It is my impression that this is a strong and valuable work. However, I believe the above comments must be addressed robustly, both in a revised manuscript and in the authors’ response, before I can recommend the paper for publication. In addition, while I find the results to be of publishable merit and quality, I believe the authors do not clearly explain the regimes in which their approach is valid and in particular, the shortcomings of the work, i.e. what further work would be required in order to obtain a well-founded set-up wherein one could properly evolve the EMRI dynamics.

Requested changes:

Comment 1) Add a comment on how they assume the validity of the point particle prescription in the full spacetime and the potential consequences of this assumption.

Response: See comment 3.

comment 2) Explicitly include the meaning of the linearised operators in equations 12 and 23

Response: We thank the Referee for their comment on Section 2.
We agree that our earlier notation for the various operators acting on the metric perturbations was somewhat imprecise. In particular, the operator acting on the (1,1) perturbation was previously written as a shorthand that implicitly included both the linear operator acting on $g^{(1,1)}$ and the source term proportional to $g^{(1,0)}g^{(0,1)}$, as later discussed in connection with the modified Regge–Wheeler and Zerilli equations.

We have now added a clarifying discussion at the end of Section 1, and modified the equations at the beginning of the following sections, to remove this ambiguity and to make explicit the separation between the linear operator acting on each perturbative order and the source terms built from lower-order fields in $q$ and $\epsilon$.

Comment 3) Add a comment on the validity of their setup from an astrophysics point of view (What can their setup actually describe?)

Response: Response to Comments 1 and 3.

We would like to thank the Referee for raising these constructive questions. We think we can answer to these points together as they touch upon complementary aspects of our setup.

Our model describes binaries with a non-rotating primary black hole embedded within a spherically symmetric matter distribution. Given the complexity of the full problem, we consider this an appropriate first step, already generalized to include an anisotropic stress-energy tensor.

Regarding the validity of our approximations, we note that the Referee’s ``thermal-mass'' criterion applies to rotation supported disks, where the vertical scale height \(H\) introduces an additional geometric scale. In our spherical configuration such a concept does not directly exist, but an analogous (Newtonian) diagnostic can be provided by the Bondi–Hoyle–Lyttleton radius of the secondary,
\begin{equation}
r_B = \frac{G m_p}{c_s^2 + v_{\rm rel}^2},
\end{equation}
which measures the region where the fluid is gravitationally bound to the small body and where nonlinear effects could, in principle, appear. Here \(c_s\) is the sound speed and \(v_{\rm rel}\) the relative velocity between the gas and the secondary.

This scale controls both the possible breakdown of the point-particle prescription (Referee’s first question) and the regime of validity of the linearized fluid equations (Referee’s second question). For orbits at radius \(r = x\,M\), one finds
\begin{equation}
\frac{r_B}{r} \sim \frac{q}{x\,[ c_s^2 + v_{\rm rel}^2 ]},
\qquad q = \frac{m_p}{M}.
\end{equation}
This ratio is naturally suppressed by the small mass ratio $q$. For a typical EMRI with \(q = 10^{-5}\), the ratio remains well below unity up to the ISCO for sound speeds or relative velocities above \(c_s,\,v_{\rm rel} \gtrsim 5\times10^{-3}c\) --- corresponding to hot or warm subsonic conditions. In this regime, \(c_s^2\) dominates the denominator of Eq. (4), so
\(r_B/r\) stays small even if the gas is nearly co-moving with the secondary. Only for very cold, nearly co-moving media ( \(c_s,\,v_{\rm rel} \lesssim 10^{-3}c\)) does \(r_B/r\) approach or exceed unity near \(x \lesssim 10\), suggesting that higher-order corrections could then become necessary.

Within our perturbative framework, characterized by \(q \ll 1\), \(\varepsilon \ll 1\), and \(r_B/r \ll 1\), the fluid remains perturbatively close to its background everywhere outside a negligible near-field region, and the point-particle description is therefore self-consistent. We emphasize, however, that the above estimates are based on Newtonian order-of-magnitude arguments. A fully relativistic matching between the inner (secondary-dominated) and outer (background+perturbation) zones would require a dedicated analysis of the local fluid flow in curved spacetime, which lies beyond the scope of the present work.

Concerning potential missing contributions to the motion of the secondary, we note that fluid elements passing within the Bondi–Hoyle–Lyttleton radius defined above are gravitationally focused by the small black hole, forming a wake that trails behind it. This overdense region can in principle exert a backreaction analogous to dynamical friction (and possibly to accretion). The strength of this interaction depends on the asymmetry of the wake and scales approximately as

\begin{equation}
F_{\rm DF} \sim \rho m_p^2\sim \mathcal{O}(q^2\,\epsilon),
\end{equation}

where $\rho$ is the background density and $\epsilon \sim \rho M^2$ in our notation. This scaling, derived from Newtonian intuition, corresponds to the same radiative order as the matter-induced corrections to the gravitational-wave fluxes discussed in the paper, but would contribute through a distinct local (near-zone) effect. A consistent evaluation of this contribution would require computing the self-consistent motion of the secondary in the perturbed geometry—i.e., feeding the $(1,1)$ metric corrections back into the worldline evolution—and thus performing a dedicated relativistic self-force analysis beyond the scope of the present work.

Finally, while our focus here is methodological and the present model still has limited astrophysical viability --- for instance due to the absence of spin and the restriction to spherically symmetric matter topologies --- it nonetheless provides a necessary first step toward a self-consistent treatment of matter-embedded compact binaries.

Interestingly, spherically symmetric configurations of black holes immersed in dense gas could, in fact, be relevant to certain recently observed compact sources - although at high-redshift - the so called ``red dots'', which may represent heavily enshrouded accreting black holes (e.g.2508.21748, 2507.09085, 2510.18301.)

We have added a discussion both in Section 1 and in the final section of the paper to clarify these points and to make more explicit the assumptions underlying the validity of our framework.

Comment 4) Address the concern regarding the exclusion of the matter fluxes in their computation. If the authors believe this inclusion is not needed, please justify why.

Response:
We thank the Referee for raising this point. In our setup, the matter distribution forms a gravitationally bound system described by a stationary, adiabatic perfect fluid with no viscosity, heat conduction, or mass outflow, while the secondary’s motion is treated at linear order in the mass ratio. Under these assumptions, the only genuinely radiative degrees of freedom are the standard gravitational-wave modes, which carry imprints of matter through coupling and background effects.

Physically, the secondary does excite fluid perturbations, but their energy is stored and redistributed within the fluid rather than transported across the external surfaces used in the flux balance law. While we believe no asymptotic matter fluxes are present, one may still expect local interactions between the perturber and the fluid (as discussed in Referee's Comment 3), which would require a dedicated SF calculation, beyond the scope of this paper.

For clarity, we have expanded the discussion in the main text where we introduce the GW fluxes, emphasizing that in our model no matter flux occurs across the horizon or at infinity. Such fluxes would require dissipative processes (viscosity, heat conduction) or mass shedding, none of which are included here.

Comment 5) In the conclusion, provide a description of the advances due to this work, as well as the limitations of this work, and discuss the road ahead to how this could actually be used for EMRI evolutions.

Response: We have expanded the Conclusions to include a discussion of the advances, limitations, and future prospects of this work, also in line with our responses to questions 1--3 above.

Comment Optional change) Improve readability and presentation of sections 2 and 3.

Response:
We have attempted to write the field equations explicitly at each order. While this is manageable for the $(0,1)$ component, the full expressions for Eqs. (25) in the revised manuscript become rather cumbersome. To maintain readability, we prefer not to include these lengthy forms in the paper, which already contains several extended expressions.

Anonymous on 2025-11-23  [id 6067]

(in reply to Sayak Datta on 2025-11-12 [id 6024])
Category:
objection
pointer to related literature

Firstly thank you to the authors for addressing my key comments. I do believe however in their response to comment four the authors come to an incorrect conclusion which must be addressed before publication.

To put it succinctly, I do not agree it is correct that the bound nature of the background fluid implies no matter fluxes as the authors say. Simply put this would imply that any bound adiabatic fluid in an astrophysical system is incapable of being ejected.

It seems to be that the authors conclusion relies on the fact that the boundary conditions present in their background solution are the same as those in the perturbative solution i.e.

"In our model the matter background is a stationary, adiabatic perfect fluid distribu-
tion with no viscosity, heat conduction, or mass outflow. With these boundary condi-
tions there is no asymptotic fluid contribution to the energy fluxes at infinity or at the
horizon. The secondary does excite fluid perturbations, but their energy is stored and
redistributed within the medium rather than transported across rh or to r →∞. "

This can not be the case, by fixing such conditions on the perturbed field, certainly at the horizon at the very least the authors are implying some form of unphysical fixed boundary conditions. As in https://arxiv.org/pdf/2307.16093, https://arxiv.org/pdf/2501.09806, and https://arxiv.org/pdf/2507.02045 the perturbed matter equations require their own dynamical boundary conditions separate form the background. It is true as the authors say that fluxes such as advected angular momentum need not necessarily arrive at infinity in the fluid sector, see section 4.1 of https://arxiv.org/pdf/1911.01428. But this suppression in the outflow is usually due to either thermodynamic or viscous processes dumping energy back into the fluid. The is seemingly the opposite of what the authors suggest.

I agree that the analysis of effects which do not arrive at the asymptotic boundaries is outside the scope of this work (as long as the shortcomings are properly noted). However it is my impression that the conclusion in response to comment four and the relevant statement added to the discussion is incorrect in their conclusion.

In the work the authors present I believe it is valid to either state explicitly that although important their work does not address the fluxes in the matter sector. An improvement upon this would be to also provide the flux formulas arising from the perturbed stress energy tensor of the fluid.

It is my opinion that in its current form the manuscript contains incorrect statements in relation to the above.

I look forward to hearing the authors response on this point. Once we have addressed this key issue I will be happy to recommend for publication.

Anonymous on 2025-12-07  [id 6114]

(in reply to Anonymous Comment on 2025-11-23 [id 6067])
Category:
answer to question

We would like to thank the referee for having accepted all our previous corrections, and for raising this further point on matter fluxes, which gives us the opportunity to clarify their role in our framework. Our original statement was not intended to suggest that a bound fluid cannot, in principle, be ejected through nonlinear processes. Our analysis concerns the linearised, adiabatic regime in which no matter flux reaches either the horizon or future null infinity. We clarify this below.

First, although bound fluids can in principle experience mass ejection through various physical mechanisms, such phenomena are inherently nonlinear and therefore lie outside the domain of validity of our linear perturbative framework as well as of the linear analyses performed in the papers cited by the referee. The perturbations induced by the small black hole in our setup do not model nonlinear ejection processes, and our conclusions concern exclusively linear matter fluxes.

To illustrate the point further, consider an idealised infinite medium. At leading order, the matter-sector perturbations contribute to the stress–energy tensor through components of the form (neglecting angular dependence on spherical harmonics for readability)
\begin{equation}
T^r{}_{t}\propto (\rho+p_r)(H_1+W)\,,
\qquad
T^\phi{}_{t}\propto (\rho+p_t)(V+U)\, .
\end{equation}
We can analyse the behaviour of matter fluxes without specifying a particular equation of state. For large radius, assuming the generic falloff (The pressures obey the same scaling, up to factors of order unity determined by the equation of state)
\[
\rho(r)\sim r^{-\beta}\qquad (r\to\infty)\, ,
\]
and recalling that \(H_1\sim r\,\partial_t\chi^{(1,0)}\sim r\) at large \(r\) (see Eq.~(88) in the paper. The same scaling applies to the velocity perturbations from the asymptotic analysis of Eq.~(70)-(71)), the components in the $T^r_t$ expression scale as
\begin{align}
T^r{}_{t}\sim r^{-\beta}r\times {\rm(constants\, coming\, from }\, H_1\, {\rm and}\, W {\rm expansion)}\ , \\
T^\phi{}_{t}\sim r^{-\beta}r\times {\rm(constants\, coming\, from}\, V\, {\rm and}\, U\, {\rm expansion)}\ .
\end{align}
The associated energy flux then behaves as
\[
\dot{E} \propto \int r^2\, T^{r}{}_{t}\, d\Omega
\sim r^{3-\beta}\, ,
\]
and analogously for the angular-momentum flux.
Thus, matter fluxes vanish at infinity provided \(\beta\gt 3\).

Importantly, this condition is not an additional assumption: it follows from the requirement of asymptotic flatness of the background (which we assume throughout the paper). Writing
\[
m(r)=4\pi\!\int_{0}^{r} \rho(r')\, r'^2 dr'\, ,
\]
asymptotic flatness implies
\[
m(r)=M + \mathcal{O}(r^{3-\beta})\, ,
\]
which requires \(\beta \gt 3\).

We emphasise, however, that this asymptotic discussion is primarily illustrative: realistic astrophysical matter distributions possess finite radial extent (or fall off rapidly beyond a finite radius), so that matter fluxes cannot reach infinity regardless of the detailed falloff. This also explains the difference with the works cited by the referee (e.g.\ arXiv:2307.16093, 2501.09806, 2507.02045), which analyse scalar fields propagating in vacuum. Scalar waves can radiate to infinity even when the background field is spatially localised, which requires dynamical boundary conditions at large radius. By contrast, fluid perturbations propagate only where the fluid is present. If \(\rho\sim r^{-\beta}\) with \(\beta\gt 3\), or if it has finite support, linear perturbations cannot produce travelling fluid waves that carry flux to the boundaries.

The situation at the horizon is even simpler. In our model we assume that the matter distribution vanishes at the horizon
(see discussion at the end of page~6), so that
\[
\rho(r_h)=p_r(r_h)=p_t(r_h)=0\, .
\]
Since the components of $T^r_t$ are proportional to \(\rho+p_r\) or \(\rho+p_t\), they vanish at the horizon, and no matter flux crosses the horizon.

In summary, the absence of matter fluxes at the horizon and at infinity follows directly from the physical properties of the background: the fluid distribution terminates before the horizon, and realistic matter configurations possess finite extent (or equivalently, a sufficiently fast falloff at large radius). Even before invoking the asymptotic analysis above, this finite support already prevents matter fluxes from reaching infinity.

We have slightly revised the text in the manuscript to clarify more explicitly why matter fluxes vanish at the horizon and at infinity within our framework.These clarifications make the underlying assumptions and physical considerations more transparent, while leaving the technical content unchanged. We have opted for a minimal modification, as additional technical detail would not improve clarity and would be beyond the scope of the point being addressed. We thank the referee for prompting this improvement and hope that the manuscript now satisfactorily addresses their concern.

---

## Round 2 · Referee Report · Dongjun Li (Referee 1) · 2025-11-21

Report

The authors have satisfactorily addressed all my comments, and I am pleased to recommend the paper for acceptance.

Recommendation

Publish (surpasses expectations and criteria for this Journal; among top 10%)

---

## Round 2 · Referee Report · Anonymous (Referee 2) · 2025-12-22

Report

Given the changes and additional comments the authors have provided I am happy to reccomend this manuscript for publication.

Recommendation

Publish (easily meets expectations and criteria for this Journal; among top 50%)

---

## Editorial Decision

in_refereeing